# Multiscale soil moisture estimates using static and roving cosmic-ray soil moisture sensors

David McJannet[1], Aaron Hawdon[2], Brett Baker[2], Luigi Renzullo[3], Ross Searle[1]

[1]CSIRO Land and Water, EcoSciences Precinct, Dutton Park, QLD, Australia

[2]CSIRO Land and Water, ATSIP, James Cook University, QLD, Australia

[3]CSIRO Land and Water, Canberra, ACT, Australia

*Correspondence to*: David McJannet (david.mcjannet@csiro.au)

**Abstract.** Soil moisture plays a critical role in land surface processes and as such there has been a recent increase in the number and resolution of satellite soil moisture observations and development of land surface process models with ever increasing resolution. Despite these developments, validation and calibration of these products has been limited because of a lack of observations at corresponding scales. A recently developed mobile soil moisture monitoring platform, known as the 'rover', offers opportunities to overcome this scale issue. This paper describes methods, results and testing of soil moisture estimates produced using rover surveys at a range of scales that are commensurate with model and satellite retrievals. Our investigation involved static cosmic ray neutron sensors and rover surveys across both broad (36 x 36 km at 9 km resolution) and intensive (10 x 10 km at 1 km resolution) scales in a cropping district in the Mallee region of Victoria, Australia. We describe approaches for converting rover survey neutron counts to soil moisture and discuss the factors controlling soil moisture variability. We use independent gravimetric and modelled soil moisture estimates collected across both space and time to validate rover soil moisture products. Measurements revealed that temporal patterns in soil moisture were preserved through time and regression modelling approaches were utilised to produce time series of property scale soil moisture which may also have application in calibration and validation studies or local farm management. Intensive scale rover surveys produced reliable soil moisture estimates at 1 km resolution while broad scale surveys produced soil moisture estimates at 9 km resolution. We conclude that the multiscale soil moisture products produced in this study are well suited to future analysis of satellite soil moisture retrievals and finer scale soil moisture models.

## 1 Introduction

Soil moisture has a strong influence of land-atmosphere interactions, hydrological processes, ecosystem functioning and agricultural productivity. The importance of this variable has led to an increase in the number and resolution of satellite soil moisture observations and the ongoing development of finer resolution land surface process models (Ochsner et al., 2013). Despite these developments, our ability to validate and/or calibrate these products is limited because of a lack of observations at matching scales. Satellite observations typically have resolutions in the order of 3 to 50 km, while broad-area modelling of soil moisture variability typically occurs at resolutions >1 km. The scale of these products are orders of magnitude larger than those of

traditional in situ sensors which creates an issue because of the well documented small scale variability in soil moisture (Vereecken et al., 2014; Western and Blöschl, 1999). Some researchers have overcome this issue by establishing soil moisture monitoring networks (Bogena et al., 2010; Smith et al., 2012), but the extent of sensor networks is still relatively small (<1 km$^2$).

More recently cosmic-ray neutron sensors (CRNS) have been deployed to provide soil moisture estimates at the hectometre scale (circular footprint, 260-600 m diameter) (Desilets and Zreda, 2013; Köhli et al., 2015). CRNS sensors measure naturally generated neutrons that are produced by cosmic rays passing through the Earth's atmosphere. Recent measurement and modelling studies (Andreasen et al., 2017a; Andreasen et al., 2017b) have shown that the CRNS sensors measure neutrons in both the thermal (<1 eV) and epithermal ranges (>1 - 1000 eV) and that sensitivities to energy range vary with environmental features present at a site (e.g. tree canopy, crop, litter). The neutron intensity above the soil surface is inversely correlated with soil moisture as it responds to the hydrogen contained in the soil and plant water and to a lesser degree to plant and soil carbon compounds (Desilets et al., 2010). The scale match between the CRNS technique and satellite observations has led to a number of recent studies which compare CRNS observations to satellite observations (Renzullo et al., 2014; Montzka et al., 2017; Kędzior and Zawadzki, 2016) and land surface models (Vinodkumar et al., 2017; Holgate et al., 2016), and use CRNS observation to parameterise models (Baatz et al., 2017; Rivera Villarreyes et al., 2014). Development of networks of CRNS across a number of countries (e.g. USA (Zreda et al., 2012), UK (Evans et al., 2016), Germany (Baatz et al., 2014), and Australia (Hawdon et al., 2014)) is providing useful time series of soil moisture information which will be valuable for years to come.

While the CRNS provides a better match to the scale of satellite retrievals and model estimates there is still a scale mismatch that prevents direct full-scale validation of these products. To address this, a mobile CRNS, called the cosmic-ray rover has been developed (Desilets et al., 2010). The rover uses the same technology as the CRNS but its design allows for mobile mapping of soil moisture across the landscape. This mobile mapping capability allows for soil moisture surveys to be undertaken over areas commensurate with satellite pixels or model domains thereby filling the gap in soil moisture observations (Chrisman and Zreda, 2013). The earliest use of the cosmic-ray rover was for repeated surveys across an area of 25 x 40 km in the Tucson Basin in order to produce a catchment scale water balance (Chrisman and Zreda, 2013). Dong et al. (2014) used a rover to map soil moisture on multiple occasions over a 16 x 10 km and a 34 x 14 km region in Oklahoma with the aim of evaluating satellite soil moisture estimates. More recently Franz et al. (2015) combined rover surveys over a 12 x 12 km area in Nebraska with CRNS measurements to develop a technique for multiscale real-time soil moisture monitoring.

This paper describes part of a research project aimed at producing soil moisture estimates at a range of scales for eventual comparison to satellite and modelled soil moisture estimates. The focus of this paper is on establishing techniques for producing spatial representations of soil moisture using CRNS sensors and a cosmic-ray rover. We will present a nested set of broad scale and intensive scale rover survey results which were collected across a 36 x 36 km area in a cropping district in Mallee region of Victoria, Australia and we will describe techniques used to convert rover measurements into soil moisture estimates using CRNS sensors and spatial soil property

information. Using statistical relationships between property scale soil moisture from rover surveys and CRNS
sensors we will present a simple approach for producing real-time property-scale soil moisture estimates in the
local area. We also use our observations at different scales to test the reliability of our experimental design.
**2 Methods**
**2.1 Site description**
The study area is located in the Shire of Buloke in the Mallee region of Victoria, Australia (Figure 1). The
measurement campaign took place across a 36 x 36 km region centred on -35.684°S, 142.858°E, which lies
between the towns of Birchip to the south and Sea Lake to the north. The Mallee is a rain fed agricultural region
with wheat and barley being widely grown. Much of the native vegetation has been removed since European
settlement. In the region of interest the landscape is flat with an elevation ranging between 50 to 120 m ASL.
The climate of the area is classified as semi-arid with an average annual rainfall of 368 mm, an average daily
minimum temperature in July of 3.6°C and an average daily maximum temperature in January of 30.7°C
(Anwar et al., 2007).
**2.2 Static cosmic-ray neutron sensors**
Cosmic-ray neutron sensors were installed at two locations in the designated field survey area (Figure 1). These
two locations are named Bishes (northern probe) and Bennetts (southern probe). Each of these sensors included
a single polyethylene shielded cosmic-ray probe (CRP-1000B, Hydroinnova, Albuquerque, USA), which
monitors neutron intensity in the epithermal to fast neutron energy range. Each system also measured barometric
pressure, temperature and relative humidity, which are required for measurement correction procedures. The
system was programmed to record data at hourly intervals and was sent via satellite telemetry (Iridium SBD
services) in near-real-time to a database on a remote server (cosmoz.csiro.au) (Hawdon et al., 2014). Prior to
deployment, the two static sensors were run side-by-side for a period of 4 days to determine if there were any
differences in counting rates that were not attributable to local conditions. Over this period the average counting
rate differed by less than 1%, thus giving confidence that differences between sensors reflect local site
characteristics alone.

In order to isolate the effect of soil moisture on neutron count measurements it is first necessary to remove
variation due to other environmental factors. The largest correction that is required is an adjustment for changes
in atmospheric pressure, but there are also corrections required for changes in atmospheric water vapor and
changes in the intensity of the incoming neutron flux. The standard correction procedures implemented across
the CosmOz network have been described in detail by Hawdon et al. (2014) therefore only a brief summary will
be provided here.

Cosmic-ray neutron intensity is particularly sensitive to elevation or the mass of air above the sensor, which is
accounted for by the by the correction factor , $f_P$ , which is defined as an exponential relationship with
barometric pressure (Zreda et al., 2008);
$$f_P = \exp\left[\beta\left(P - P_{ref}\right)\right]$$ Eq. 1
where $P$ is atmospheric pressure (mb) and $P_{ref}$ is the reference atmospheric pressure (mb); which is calculated
using standard formulas based on site elevation (NASA, 1976). The atmospheric attenuation coefficient ($\beta$,
$cm^2$ $g^{-1}$ or $mb^{-1}$) for neutron-generating cosmic rays has been calculated for each of our sites using the method
described by Desilets et al. (2006).

Water vapor in the atmosphere has the same neutron moderating capacity as water in the soil and as such will
influence the total neutron count (Zreda et al., 2012). A correction factor for atmospheric water vapor effects
was developed by Rosolem et al. (2013) and it utilises near surface absolute humidity ($\rho_{v0}$, g $m^{-3}$), which is
derived from measurements of temperature, atmospheric pressure and humidity. The correction factor for
atmospheric water vapor ($f_{wv}$) is derived from;
$$f_{wv} = 1 + 0.0054\left(\rho_{v0} - \rho_{v0}^{ref}\right)$$ Eq. 2
where $\rho_{v0}^{ref}$ is the reference absolute humidity, which we set to 0 g $m^{-3}$ (i.e. dry air).
To account for variations in incoming neutron flux an intensity correction factor is calculated by normalising the
source intensity to a fixed point in time (Zreda et al., 2012). The correction factor for incoming neutron intensity
($f_i$) is expressed as;
$$f_i = \frac{I_m}{I_{ref}}$$ Eq. 3
where $I_m$ is the selected neutron monitor counting rate at any particular point in time and $I_{ref}$ is a reference
counting rate for the same neutron monitor from an arbitrary fixed point in time which is 1 May 2011. Neutron
monitor data is sourced from the Neutron Monitor Database (NMDB; www.nmdb.eu). Both of these sites utilise
data from the Lomnický štít Observatory in Slovakia.

The counting rate is also scaled to sea level and high latitude to enable comparison between sensors. Scaling
factors for converting counting rate to sea level ($f_s$) and high latitude ($f_l$) are described by Desilets and Zreda
(2003) and Desilets et al. (2006).

Final corrected counts ($N$) are calculated using the following equation;
$$N = N_{raw}\left(\frac{f_P f_{wv}}{f_i}\right)\left(\frac{f_s}{f_l}\right)$$ Eq. 4
Where $N_{raw}$ is the uncorrected neutron count from the CRP. Corrected neutron counts were converted to
volumetric soil moisture content ($\theta$) using the calibration function generated by Desilets et al. (2010) and
modified by Bogena et al. (2013):
$$\theta = \left( \frac{0.0808}{\left( \dfrac{N}{N_0} \right) - 0.372} - 0.115 - w_{lat} - w_{SOM} \right) \rho_{bd}$$
Eq. 5

where $N_0$ is the neutron intensity in air above a dry soil which is obtained from field calibration, $w_{lat}$ is lattice
water content of the soil, $w_{SOM}$ is soil organic matter expressed as a water equivalent (see below), and $\rho_{bd}$ is bulk
density of the soil.

Field calibration at each site involved collection of gravimetric and volumetric soil samples at three distances
from the probe (25m, 100m and 200m) along each cardinal and inter-cardinal direction (i.e. 8 radial directions).
At each sample point, soil cores were taken to calculate volumetric soil moisture content for three depths (0 to 5
cm, 10 to 15 cm, and 25 to 30 cm), giving a total of 72 samples per calibration. Water content from samples was
determined by drying samples at 105°C for 24 hours (Klute, 1986). The depth weighted soil moisture from field
calibration was calculated using the method proposed by (Franz et al., 2012) and corresponding corrected
neutron count is used to determine $N_0$ in Eq. 5. Hydrogen held within the lattice structure of the soil minerals
and organic material can also effect neutron count rate and, hence, need to be considered in calculation
procedures. Lattice water ( $w_{lat}$ ) was determined from the amount of water released at 1000°C preceded by
drying at 105°C. Soil organic carbon was estimated by measuring total organic carbon in samples using Heanes
wet oxidation, method 6B1 in Rayment and Higginson (1992). Following Franz et al. (2013) and Bogena et al.
(2013), the organic carbon was assumed to be present as cellulose, $C_6H_{10}O_5$, and this was converted into an
equivalent amount of water ( $w_{SOM}$ ) by multiplying measured soil organic carbon by 0.556, which is the ratio of
five times the molecular weight of water to the molecular weight of cellulose.
**2.3   Rover system**
The rover system is based around a set of 16 custom made tube capsules supplied by Hydroinnova
(Albuquerque, USA), which are similar to those used for the static cosmic-ray neutron sensors but larger. The
rover has counting rates approximately18 times greater than that of a standard static sensor under the same
condition, thus, allowing for measurements to be made at one minute intervals. For a volumetric soil moisture
content of 10% a count rate of around 350 c min$^{-1}$ was recorded. The set of 16 tubes is mounted in a trailer from
which additional measurements of air temperature, relative humidity, atmospheric pressure and location were
also made. Pictures of the rover system are available on the CosmOz webpage (http://cosmoz.csiro.au/about-
cosmoz/). While mobile, the measurements from the system were monitored in real-time on a screen in the cabin
of the tow vehicle. A dash mounted camera was also used to collect images at one minute intervals during the
survey.

For this investigation a nested design of broad scale and intensive localised measurements was implemented.
The broad scale design included a survey over an area with dimensions of approximately 36 x 36 km which
encapsulated a single Soil Moisture Active Passive (SMAP) satellite pixel. Using typical counting rates for this
area and by targeting an output resolution for soil moisture of 9 x 9 km we calculated that the maximum driving
speed for this survey was 90 km h$^{-1}$. This provided a good density of measurement points for interpolation
purposes. The survey area and measurement points from the driving track are shown in Figure 2. The broad
scale surveys typically took 10 h to complete, involved over 600 measurements and the average speed travelled
was around 60 km h$^{-1}$. The intensive scale survey covered an area of approximately 10 x 10 km and was located
in the south eastern corner of the broad scale survey (Figure 2). In this survey a target resolution for soil
moisture of 1 x 1 km was used for which we calculated that the maximum driving speed should not exceed 30
km h$^{-1}$. Much of the driving for the intensive scale surveys was around field boundaries and on unsealed roads.
At 1 km resolution the intensive scale survey results were well matched to farm property scale in this region.
Intensive scale surveys also took approximately 10 h to complete with more than 600 measurement point being
collected. The average speed during these surveys was 20 km h$^{-1}$. Survey tracks were defined for both surveys
prior to undertaking measurement using maps of the local road network. These maps were loaded into GIS
software and were used to guide navigation on each survey run.

The nested design of the intensive and broad scale surveys (Figure 2) enables the accuracy of broad scale survey
estimates to be assessed. To undertake such an analysis we selected a 9 x 9 km area within the area of survey
overlap (Figure 2) and derived corresponding soil moisture at resolutions of 1, 3 and 9 km. In such an analysis
the intensive survey results are considered as a point of truth for broad survey results.

As well as enabling production of direct farm property-scale estimates at the time of the surveys, the intensive
scale survey results were used to derive a much higher time resolution soil moisture product at the property
scale. This was achieved using spatial regression analysis with the continuous soil moisture measurements at the
static CRNS observations at Bennetts. Linear regression equations were derived for each property by comparing
the soil moisture content at the Bennetts CRNS versus the corresponding rover survey soil moisture for each
property in turn. Using this approach, regression relationships were developed between the Bennetts CRNS and
50 properties identified within the intensive survey area for the three surveys undertaken. These relationships
enable production of continuous farm property scale in this area. This approach assumes that rainfall is
relatively uniform across the region and that crops are planted across all periods; both of which are typical in
this study area.

Procedures used for correcting static cosmic-ray neutron sensor counts (Eq. 1 to Eq. 4) were also applied to the
rover data. Continually varying elevation, location, pressure, temperature and humidity were used for these
calculations. Soil moisture was also calculated in the same way as for the static sensors (Eq. 5) but there was a
requirement for spatial information regarding bulk density, soil organic matter and lattice water content. The
Soil and Landscape Grid of Australia provides ~90 x ~90 m pixels of digital soil attributes including bulk
density (Viscarra Rossel et al., 2014a) and soil organic carbon (Viscarra Rossel et al., 2014b)  at depths of 0-5
cm, 5-15 cm and 15-30 cm which are useful for applying to rover surveys. The Soil and Landscape Grid of
Australia does not provide any lattice water information but it does provide information on clay content
(Viscarra Rossel et al., 2014c) and others (Greacen, 1981; Avery et al., 2016) have shown that clay content is
often a good predictor of lattice water. In this study we investigated whether such a relationship existing for the
soils in the study area. To do this we collected 36 samples for lattice water analysis; this included 25 distributed
samples in the broad scale survey area, 9 samples across the intensive scale survey area and the 2 samples
collected as part of the calibration of the static probes. These samples were from cores extracted from 0-30 cm
depth. The spatial maps of bulk density, clay content and organic carbon used in the rover calculation
procedures are shown in Figure 3, also shown for site characterisation is the digital elevation model for the
survey area.

Use of Eq. 5 in rover surveys also requires specification of a suitable $N_0$ value. For the static sensors this value is
derived through the calibration procedures. To calculate $N_0$ for the rover we undertook side-by-side comparisons
with the static sensors which involved parking next to a static sensor for 12 hours prior to a survey. The average
counts from the rover and static sensor were then compared to derive a suitable scaling approach to derive a
rover-specific $N_0$. Similar cosmic-ray neutron sensor cross-calibrations were undertaken by (Baatz et al., 2015)
to account for sensor specific differences. Both broad scale and intensive scale surveys were undertaken on three
separate occasions on consecutive days during April 2016, June 2016 and March 2017.

Interpolation of the rover count data was required to produce a spatial representation of count rates for the entire
survey area. To achieve this the Variogram Estimation and Spatial Prediction with Error (VESPER) software
package (Minasny et al., 2005) was used. VESPER was used to undertake conventional kriging with a global
variogram. An exponential variogram model was used for both survey scales and an interpolated grid of
corrected rover count rate was produced at 90 m resolution to match that of the underlying soils information.

**2.4    Comparison data sets**
Two independent datasets were utilised for comparison to soil moisture estimates from our rover surveys; 1)
opportunistic point samples collected during each survey, and 2) modelled soil moisture estimates from the
Australian Bureau of Meteorology's Australian Water Resources Assessment Landscape model, known as
AWRA-L.

Soil samples were collected at approximate predefined points, as shown in Figure 2, during each of the rover
surveys. A full set of samples was collected during the April 16 surveys and smaller sub-sets were collected
during the later surveys. At each sampling location a single 0 - 30 cm core was extracted. Gravimetric water
content for these cores was determined by drying samples at 105°C for 24 hours. For comparison purposes,
rover volumetric soil moisture estimates for the nearest pixel (9 km resolution for broad scale and 1 km
resolution for intensive) were extracted and divided by the corresponding average bulk density for that pixel to
produce an equivalent gravimetric estimate of soil moisture. We note here that there is a large scale discrepancy
between these datasets and highlight that the point samples only offer an approximate guide as to the accuracy
of rover survey results.

AWRA-L is a daily 0.05° (~5km) grid based, distributed water balance model. It simulates the flow of water
through the landscape with rainfall entering the grid cell through the vegetation and soil moisture stores and
leaving the grid cell through evapotranspiration, runoff or deep drainage to the groundwater. The
implementation and testing of the AWRA-L model has been described by numerous authors (Wallace et al.,
2013; Van Dijk, 2010; Viney et al., 2014). Of particular interest to this study is the AWRA-L estimate of root
zone soil moisture which covers a depth of 0 - 100 cm. The root zone represents a deeper soil zone than the
effective depth of the rover but provides our best source of comparison data. When comparing 5 km resolution
AWRA-L soil moisture estimates to those from the 9 km resolution broad scale rover survey the nearest
AWRA-L pixel to the 9km pixel centroid was used. When comparing the AWRA-L soil moisture to the 1 km
resolution intensive scale survey the intensive scale pixels were grouped to produce a corresponding 5 km
resolution product. AWRA-L soil moisture was reported in percentage capacity between 0 - 100% while the
rover results were in volumetric units, no attempt was made to convert between units and the comparison
focused on the strength of the fit between the data sets.
**3    Results**
**3.1   Static CRNS calibration**
Calibration of the two CRNS occurred under different soil moisture conditions; at Bennetts the depth weighted
soil moisture content was 0.13 $m^3$ $m^{-3}$, while at Bishes it was 0.08 $m^3$ $m^{-3}$. Fitting of the calibration curve to
these two sites (Figure 4) resulted in very similar dry soil ($N_0$) counting rates with analysis of the data collected
at Bennetts producing an $N_0$ of 1541 c $h^{-1}$ and that from Bishes producing an $N_0$ of 1583 c $h^{-1}$. Across the soil
moisture range of 0 to 0.5 $m^3$ $m^{-3}$ the average soil moisture difference between the two curves in Figure 4 was
0.019 $m^3$ $m^{-3}$. These differences are very small and reflect the fact that hydrogen represented by the biomass
pool is basically non-existent at these sites.
**3.2   Rover calibration**
Calibration of the rover was undertaken through side-by-side comparison with the Bennetts CRNS and the
Bishes CRNS on two separate occasions each. These comparisons covered a range of soil moisture conditions
over four separate 12 h periods. Table 1 shows the corresponding neutron count rate for the rover and each
CRNS and the scaling factor that converts static CRNS counting rate to a rover equivalent; this scaling factor is
used to scale the $N_0$ values derived for each static sensor to an equivalent $N_0$ for the rover. Despite the
differences in conditions and site characteristics, the scaling factor remained relatively constant, as did the
derived $N_0$ for each comparison period. Given the relatively constant relationship between the rover and static
sensors an average $N_0$ of 460 c $min^{-1}$ was derived and this value was applied across all surveys.
**3.3   Spatial lattice water information**
A comparison of clay content and lattice water content for 36 spatially distributed samples shows a strong linear
relationship ($R^2$ =0.7) across a broad range of clay content (4–56%) (Figure 5). This relationship was applied to
the spatial clay content data set from the Soil and Landscape Grid of Australia (Viscarra Rossel et al., 2014c) to
produce an equivalent lattice water dataset at 90 m resolution which was utilised in rover surveys.
**3.4  Spatial estimation**
Example variograms from the kriging procedures used for broad scale and intensive surveys are shown in Figure
6. Both surveys utilise exponential variogram models however the fit is different with the intensive scale
surveys having a distinct 'sill' and broad scale variograms showing no 'sill' at all. The 'sill' in a variogram
represents the value at which the fitted model levels out (see Figure 6). The presence of a sill indicates that there
is a distance (known as the 'range') between pairs of points beyond which there is no spatial correlation. The
range is important as it is related to the spatial scale of the variability in neutron intensity. The lack of a sill for
the broad scale survey reflects differences in variability in neutron observations at this larger scale. The
variogram model for the intensive surveys showed more cyclicity (or 'hole effect') which could be related to
underlying geological periodicity (Yang and Kaleita, 2007). The empirical variograms were well described by
the exponential models giving confidence in interpolated rover counts across the respective survey areas.
**3.5  Intensive scale rover surveys**
Interpolated counts and derived volumetric soil moisture content for each of the three intensive scale surveys is
shown in Figure 7. A large range in soil moisture content was observed over the three surveys with values
ranging between 0.01 $m^3$ $m^{-3}$ in April 2016 through to 0.30 $m^3$ $m^{-3}$ in June 16. Higher than average counting
rates and, hence, lower soil moisture were consistently observed in the central northern region of the survey
area. This area is characterised by a ridge of sandy soil with rock fragments and is known locally as 'Sandhill'.
Wetter soil moisture conditions were observed through the central and southern parts of the survey area.

Comparison of intensive rover survey soil moisture estimates for the CRNS locations at the three different
survey dates shows excellent agreement between the two measurement methods (Figure 8). The rover survey
estimate is taken from the 1 km resolution soil moisture estimate for the corresponding CRNS pixel.
Comparisons of estimates for the Bennetts CRNS shows differences of less than 0.025 $m^3$ $m^{-3}$ for all three
occasions. The rover survey estimates tended to underestimate the soil moisture measured at the Bishes CRNS.
The largest difference was during the April 2016 survey where soil moisture was underestimated by 0.04 $m^3$ $m^{-}$
$^3$. It is possible that this underestimation is a result of local interpolation issues. The Bishes CRNS is in close
proximity to the sandy ridge known as 'Sandhill' which represents a distinct zone of low soil moisture (Figure
7). The effect of this abrupt change is likely to be 'smoothed' within the area that also encompasses the Bishes
CRNS.

Figure 9a shows a comparison of rover gravimetric soil moisture against corresponding soil moisture from the
grab samples collected during each survey. The comparison shows strong correlation ($R^2 = 0.80$) and data points
are scattered around the 1:1 line. There is more scatter observed in the data under wetter conditions but this is
likely to be related to a greater relative difference in spatial soil moisture following rainfall events. Similarly, the
comparison of rover volumetric soil moisture against modelled root zone soil moisture from the AWRA-L
model (Figure 9b) also shows good correlation ($R^2 = 0.79$). This comparison is complicated by the fact that the
rover estimate represents an effective measurement depth of between 10 to 25 cm while the root zone soil
moisture is an estimate between 0 and 100 cm, despite this the agreement is still good. Comparison to these two
independent soil moisture products with the rover surveys increases confidence in rover survey results at the
intensive scale.

The rover surveys at the intensive scale also offer the opportunity to estimate soil moisture at the farm property
scale. A number of properties in the intensive scale zone are identified in Figure 10 and the intensive scale rover
survey from March 2017 has been used to derive property average soil moisture conditions in this figure. The
average size of the identified properties is approximately 1 km$^2$.

Point-to-area linear regression modelling based on continuous CRNS measurements from the Bennetts sensor
and three intensive rover surveys was applied to 50 properties identified in the intensive survey area  and very
strong linear relationships were derived with an average $R^2$ value of 0.97 (range = 0.87-1.00, see Table A1 for
full results). We note here that only three surveys were available for developing these relationship and further
surveys and cross validation is recommended for future work. Application of these regression models to derive
time-series of property scale soil moisture for three example properties is given in Figure 11.

**3.6    Broad scale rover surveys**
Interpolated counts and derived volumetric soil moisture content for each of the three broad scale surveys is
shown in Figure 12. The common feature of all of the survey dates is the tendency for higher counts and, hence,
lower soil moisture to occur at the north-western region of the survey area and lower counts and, hence higher
soil moisture to occur in the south-eastern region. These patterns reflect soil textures in the region with sandier
soils and dunes with low clay content in the north-western and higher clay content soils in south-east. The driest
soil moisture conditions were experienced during the April 2016 survey with a mean soil moisture of 0.05 m$^3$ m$^-$
$^3$ (range = 0.01–0.10 m$^3$ m$^{-3}$) and the wettest were observed during the June 2016 survey with a mean soil
moisture of 0.17 m$^3$ m$^{-3}$ (range = 0.09–0.27 m$^3$ m$^{-3}$). The March 2017 survey provided intermediate soil
moisture conditions with a mean for the region of 0.09 m$^3$ m$^{-3}$ (range = 0.04–0.15 m$^3$ m$^{-3}$).

Figure 13a shows a comparison of rover gravimetric soil moisture against corresponding soil moisture from the
grab samples collected during each survey. The comparison shows reasonable correlation ($R^2 = 0.64$) and data
points tend to be scattered around the 1:1 line. Given the scale difference between these products (9 km vs point
sample) the observed scatter is not surprising. Figure 13b shows a comparison of rover volumetric soil moisture
against modelled root zone soil moisture from the AWRA-L model. The closer scale match between these two
products (9 km vs 5 km) when compared to the point samples, results in a much higher correlation between the
two data sets ($R^2 = 0.78$). As with the intensive survey comparison interpretation of the results is complicated
because the measurement depth of the rover (10 to 25 cm) is much less than the AWRA-L root zone soil
moisture (0 and 100 cm). Despite these differences the two products are still remarkably well correlated and the
good agreement between the rover estimates and the AWRA-L estimates, both spatially and across a range of
soil moisture conditions, provides further evidence that the rover experimental design and data processing
procedures are reliable.

Broad scale survey soil moisture estimates were also tested by comparison with intensive survey results at scales
of 1, 3 and 9 km in an overlapping 9 x 9 km region (Figure 2). The difference in soil moisture estimates between
the broad and intensive scale surveys for different resolutions on each of the three survey dates is shown in
Figure 14. The broad scale survey estimates are clearly not a good representation of 1 x 1 km scale soil moisture
as survey speeds and sampling points are not detailed enough to pick up local soil moisture variations at current
counting rates. Differences of up to $\pm 0.10$ m$^3$ m$^{-3}$ were observed. At 3 x 3 km resolution the performance of the
broad scale survey estimates improves but there are still some distinct zones where soil moisture differed by as
much as $\pm 0.06$ m$^3$ m$^{-3}$. At the 9 x 9 km scale, for which the broad scale surveys were designed, differences in
soil moisture between the intensive and broad scale surveys was minimal. On all three occasions the difference
was less than 0.005 m$^3$ m$^{-3}$. These comparisons validate our broad scale experimental design and give
confidence in the 9 x 9 km resolution soil moisture produced from our rover surveys.
**4    Discussion**
Static CRNS calibration at Bishes and Bennetts produced very similar dry soil counting rate ($N_0$). This similarity
has resulted because hydrogen in soil water, lattice water and organic matter is accounted for in the calibration
process and because both sites are devoid of above ground biomass. The effect of biomass on $N_0$ has been noted
by Hawdon et al. (2014) who compared $N_0$ values from eight probes from across the Australian CRNS network
with site biomass and also by Baatz et al. (2015) who proposed an empirical biomass correction for CRNS
calibration. This finding has important implications for rover surveys in this region as the landscape in the
Mallee region is almost entirely cleared of forest and above ground biomass is represented by pasture and crop
cover.  McJannet et al. (2014) calculated that pasture represented a biomass water equivalent of just 0.6 mm a
value similar to that derived by Baatz et al. (2015) for areas dominated by crops; these small values show that
these small hydrogen pools will to have little impact on neutron counts (McJannet et al., 2014).

In this present study the $N_0$ value for converting rover neutron counting rates to soil moisture content was
derived through side by side comparison with the two CRNS sensors. A similar approach was employed by
Chrisman and Zreda (2013) using a single CRNS as a reference point and by Dong et al. (2014) using a network
of in situ measurements. Rover surveys undertaken by Franz et al. (2015) also used comparison with static
CRNS sensors but in their investigations a further correction was introduced to account for variations in above
ground biomass. Locations with greater biomass should adopt a calibration schemes that include this hydrogen
pool (i.e. Baatz et al., 2015; Franz et al., 2013).

Rover surveys require information on the spatial variation in bulk density, soil organic matter and lattice water
for calculation of soil moisture content using conventional approaches. While pre-existing bulk density and
organic matter datasets exist for Australia we had to derive a lattice water dataset based on a strong region-wide
relationship with clay content. The relationship we derived for the study area was different to that proposed by
Greacen (1981) for Australian soils and may reflect differences in the soil types included in the analysis. With
the intent of producing a similar spatial lattice water dataset for the continental United States, Avery et al.
(2016) derived relationships with clay content but found that relationships were weak for many soil taxonomic
group. For best local results a spatial sampling such as that utilised in this present study is recommended.

A factor that has not been accounted for in our rover surveys is the potential impacts of roads on our survey
results. By design roads will have a low moisture content and the impact of this narrow strip within the sensor
footprint on survey results has not yet been accounted for in any operational rover studies reported in the
literature.  Using neutron modelling approaches Köhli et al. (2015) demonstrated that a CRNS is most sensitive
to soil moisture in the nearest tens of metres and showed that dry roads can contribute to an over estimate of
neutron counts by a few percent. The dry roads will be over-represented in the measured neutron intensity as the
sensitivity of neutron intensity to hydrogen is greater at the dry end of the scale (Andreasen et al., 2017a). A
more recent study by Schrön et al. (In Review) using neutron transport simulations and dedicated field
experiments supports the findings of Köhli et al. (2015). Schrön et al. (In Review) found that the effects of roads
are greatest when surrounding soil moisture is much higher than road moisture content. In the survey areas in
which our broad scale rover surveys were undertaken more than 70% of the roads were unsealed and many of
the sealed roads were only one lane wide; while this does not remove the issue it does lessen the potential
impact on reported results considerably. The impact of roads on our intensive scale surveys is likely to be even
less as 60% of the observations were made while driving around property boundaries (i.e. not properly formed
roads) and a further 30% were on unsealed roads. While the impact of roads may not be a major issue for the
present study it is an issue that needs some warrants consideration in future surveys.

Intensive scale surveys were designed to produce a 1 x 1 km resolution soil moisture product and comparison to
static CRNS observations, spatially distributed point samples and AWRA-L model predictions support this.
While the point samples and model estimates cannot be considered the 'truth' they do provide a good guide as to
rover performance and the agreement with these estimates provides confidence in intensive scale rover results.
Detailed soil moisture maps highlight the impact that soil properties have on observed soil moisture with sandier
locations being typically drier when compared to those with more clay. Property scale soil moisture estimates
led to the development of point-to-area style regression models which then enabled continuous estimates of soil
moisture to be made at the property scale. Property-scale regression models were strong but it is noted that these
are based on data from three surveys. A more thorough investigation is recommended and this should include
further surveys and cross validation experiments. The opportunity also exists to use similar point-to-area scaling
techniques to derive high temporal resolution soil moisture products at other set resolutions (e.g. 1 km) which
would make for ideal datasets for testing model and satellite soil moisture estimates. The regression modelling
undertaken showed that temporal patterns in soil moisture were strong. Similar observations have been reported
for other studies (Kachanoski and Jong, 1988; Grayson and Western, 1998; Vachaud et al., 1985). According to
Yang and Kaleita (2007) spatial patterns of soil moisture exhibit some degree  of temporal stability which is
related to time invariant attributes such as topography and soil characteristics. With the relatively flat
topography in Mallee study area and the assumption that rainfall inputs and crop growth are similar between
properties, it is likely that differences in the slopes and intercepts of relationship between CRNS observations
and property scale soil moisture (see Table A1) are being controlled by local soil characteristics. Changes in
local crops and local scale differences in rainfall inputs (i.e. small convective storms) do of course have the
potential to change these point-to-area relationships but if these factors can be accounted for then useful spatial
and temporal soil moisture datasets can be produced.
Comparison of broad scale rover soil moisture estimates against those from point samples and the AWRA-L
model showed good agreement across both space and time, thus providing further evidence that the rover
experimental design and data processing procedures were reliable. Agreement between rover estimates and
model estimates was particularly good and this reflects the closer match in scale of these two products.
Comparison with emerging satellite, measurement, and modelled soil moisture products will help to further
assess rover approaches and results in the future. Broad scale surveys produced reliable soil moisture estimates
at 9 x 9 km resolution although the faster survey speeds and lower measurement density meant that this survey
was unable to distinguish many of the smaller scale soil moisture variations revealed at the finer resolution and
slower survey speeds of the intensive scale survey. This clearly supports the need to design rover surveys for the
scale of analysis to be eventually undertaken.
**5    Conclusion**
In this study we presented an investigation designed to produce soil moisture estimates across a range of scales.
Our investigation involved static CRNS sensors and rover surveys at both broad and intensive scales. We
established techniques for converting neutron counting rates from the rover to soil moisture using side-by-side
comparisons with static CRNS sensors and spatial datasets of soil characteristics. In particular we found that
lattice water was strongly related to clay content in the study area and used this relationship to derive a spatial
representation of lattice water.
Rover surveys were undertaken across soils ranging in moisture content from 0.01 to 0.30 $m^3$ $m^{-3}$ and
comparison with spatial distributed point samples and model estimates showed that reliable results were
produced across all conditions. The slower driving speeds and denser sampling network of the intensive surveys
provided representation of local soil moisture variations at resolutions down to 1 x 1 km. Stability in observed
spatial patterns of soil moisture were used in a regression modelling approach to produce time series of property
scale soil moisture based on CRNS observations. Broad scale surveys, which incorporated higher driving speeds
and sparser sampling points, were shown to produce excellent representations of soil moisture at 9 x 9 km pixel
resolution making them well suited for assessing variation in this parameter at a regional scale. The multiscale
application of the rover makes it a unique tool for addressing soil moisture questions across scales previously
not possible. The multiscale soil moisture products produced in this study are well suited to future analysis of
both satellite soil moisture retrievals and finer scale soil moisture models.

**Acknowledgements**
The authors acknowledge the cooperation of Tim McClelland and family who allowed access to their properties
for installations and surveys. We are grateful to staff from the Birchip Cropping Group who helped with surveys
and sample analysis. Three anonymous reviewers, Roland Baatz and Auro Almeida are thanked for their
valuable review comments. Funding for this research was provided by the Department of Agriculture & Water
(Grant Agreement GMS-2582) and CSIRO. We acknowledge the NMDB database (www.nmdb.eu), founded
under the European Union's FP7 programme (contract no. 213007) for providing neutron monitor data.
Lomnický štít neutron monitor data (LMKS) were kindly provided by the Department of Space Physics,
Institute of Experimental Physics, Košice, Slovakia. AWRA-L soil moisture estimates were provided by the
Australian Bureau of Meteorology landscape water balance modelling program
(www.bom.gov.au/water/landscape).

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

**6    Tables and captions**

**Table 1. Side-by-side comparison of average neutron counts for the static CRNS's (Bishes and Bennetts) and the**
**rover for 4 different 12 hour periods. Also shown are the average soil moisture values for each date, static CRP to**
**rover scaling factors and derived dry soil counting rate, $N_0$, for the rover. All counts are in c min-1 for application to**
**rover data.**

| Date | Site | Static CRNS average counts (c min$^{-1}$) | Static CRNS average soil moisture (m$^3$ m$^{-3}$) | Rover average counts (c min$^{-1}$) | Static to rover scaling factor | Static CRNS $N_0$ (c min$^{-1}$) | Derived rover $N_0$ (c min$^{-1}$) |
|---|---|---|---|---|---|---|---|
| 10 April 2016 | Bishes | 21.74 | 0.08 | 370.0 | 17.0 | 26.4 | 449 |
| 1 March 2017 | Bishes | 20.4 | 0.10 | 364.8 | 17.9 | 26.4 | 471 |
| 9 June 2016 | Bennetts | 15.23 | 0.28 | 268.1 | 17.6 | 25.7 | 452 |
| 2 March 2017 | Bennetts | 16.8 | 0.16 | 307.6 | 16.8 | 25.7 | 469 |
|  |  |  |  | **Average** | **17.3** | **Average** | **460** |




    **7    Figures and captions**



**Figure 1. Location of field site in western Victoria, Australia. Yellow rectangle shows extent of broad scale rover surveys (36 x 36 km) and red rectangle shows extent of intensive surveys (10 x 10 km). Blue and red stars indicate the location of the Bishes and Bennetts cosmic-ray neutron sensors. Imagery data: Google, TerraMetrics 2017.**




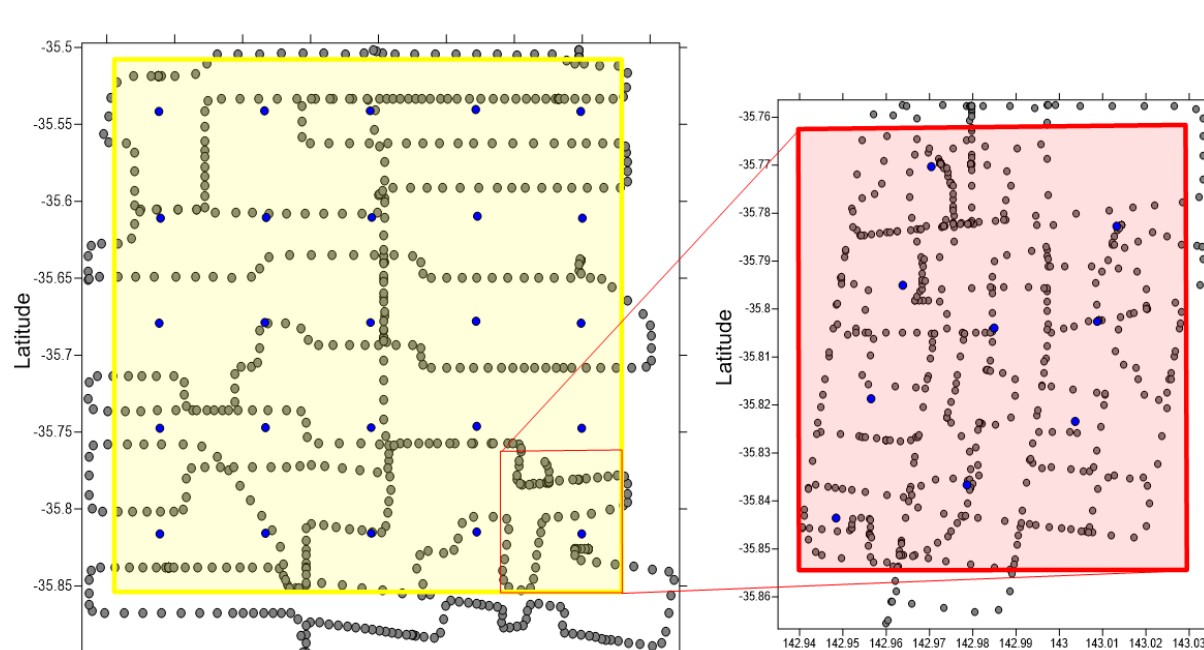


**Figure 2. Rover survey extents and sampling points for the broad scale and intensive scale measurement campaigns.**
**Sampling points from April 2016. The yellow box (~36km x 36km) delineates the broad scale survey extent and the**
**red box (~10km x 10 km) delineates the intensive scale survey extent. Blue points in each figure represent**
**approximate locations of gravimetric soil moisture sampling points.**


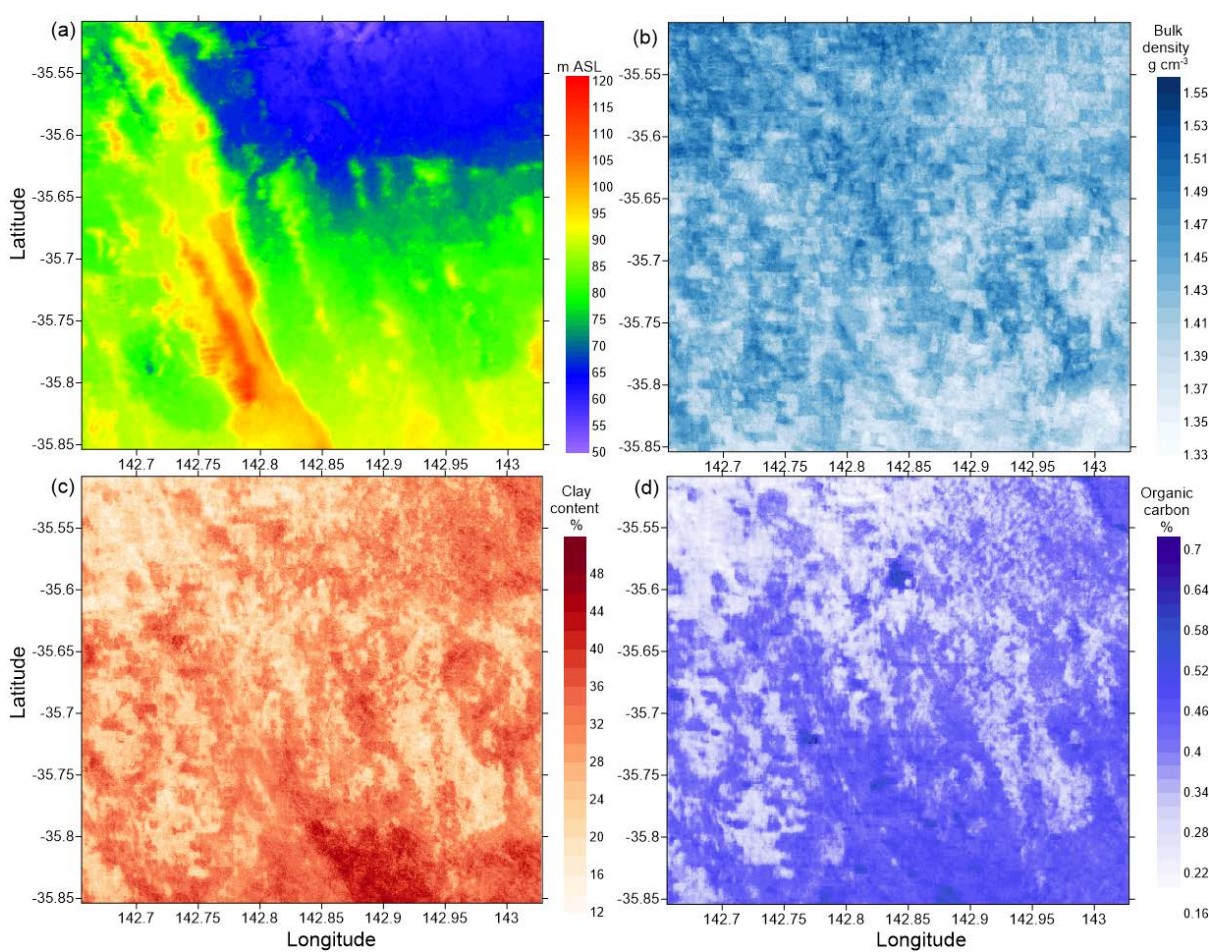


**Figure 3. Field survey area DEM (a), depth weighted 0–30 cm bulk density (b), depth weighted 0–30 cm clay content (c), and depth weighted 0–30 cm organic matter content (d).**


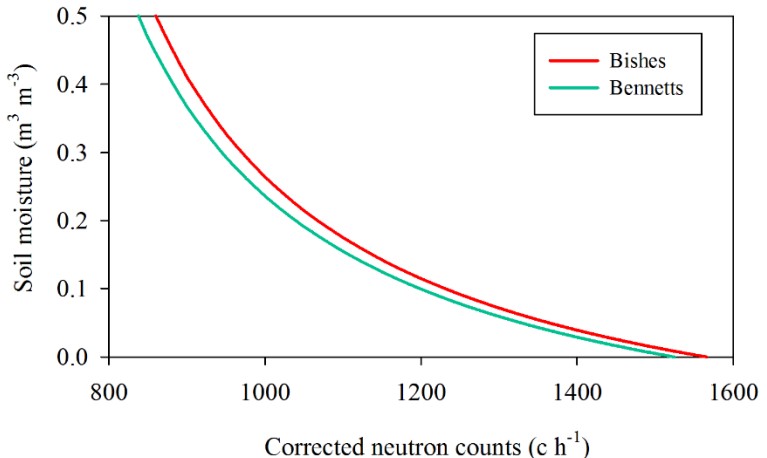


**Figure 4. Calibration curves for converting corrected neutron counts to soil moisture content for the Bishes and Bennetts cosmic ray soil moisture sensors. The dry soil counting rate, $N_0$, is 1583 c h$^{-1}$ for Bishes and 1541 c h$^{-1}$ for Bennetts.**



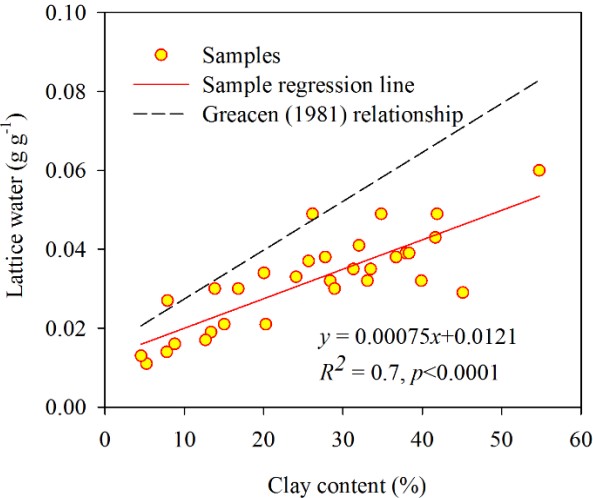


**Figure 5. Clay content vs Lattice water showing sample points from the study area and fitted relationship. Also shown for reference is the relationship proposed by Greacen (1981).**



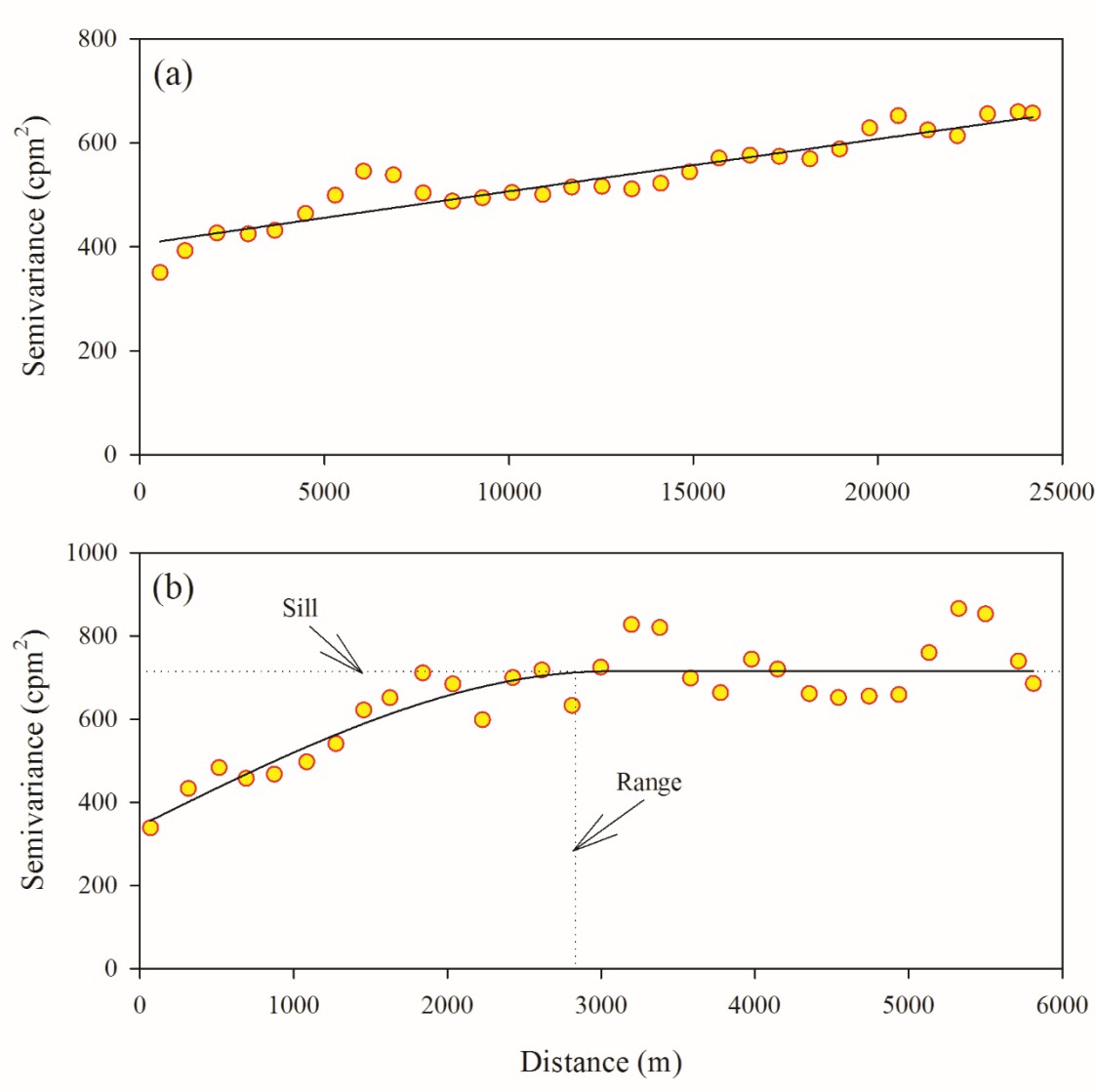


**Figure 6. Example variograms used for block kriging for broad scale and intensive surveys. The broad scale variogram is from April 2016 (a) and the intensive scale variogram is from June 2016 (b). The sill and the range are shown in (b).**



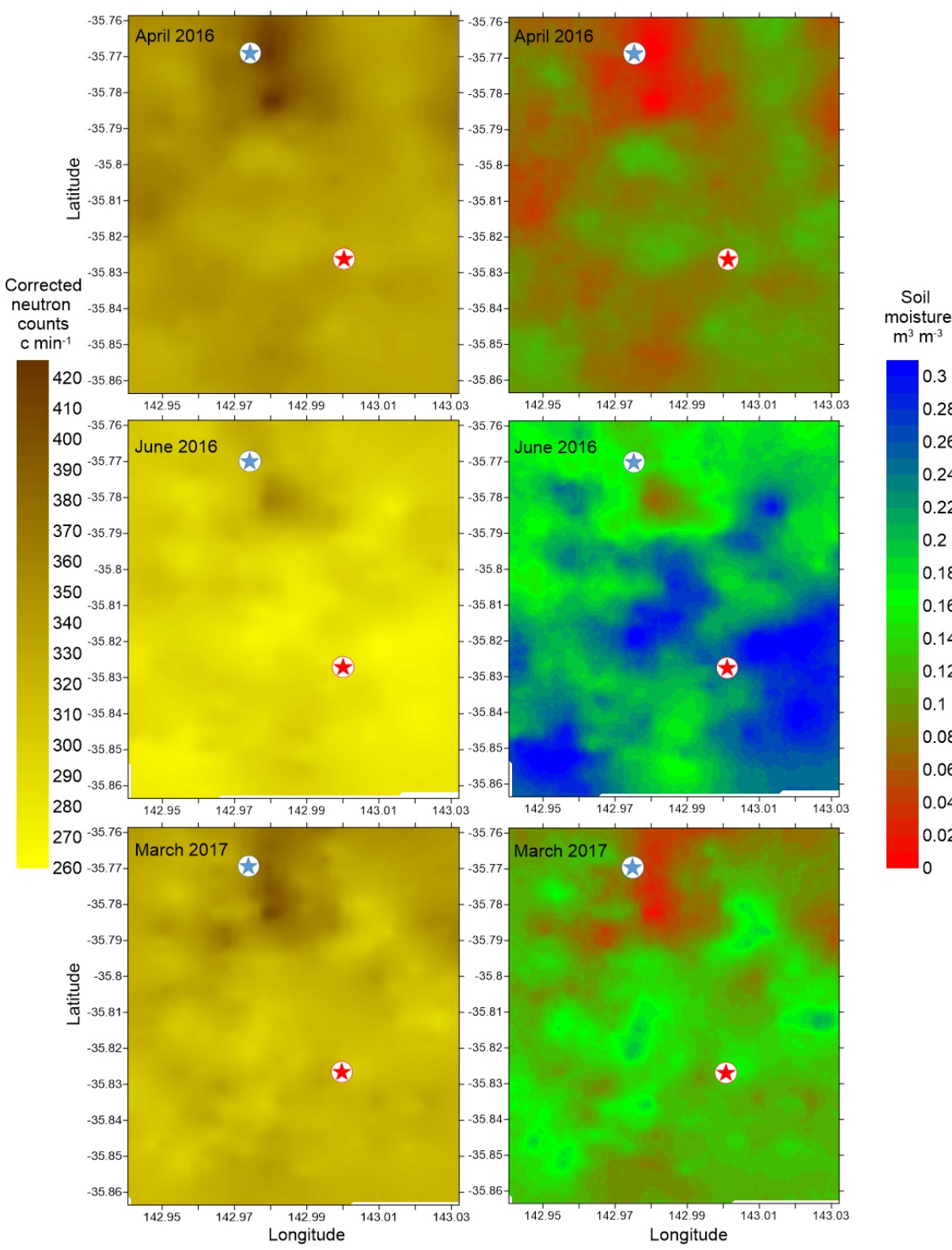


**Figure 7. Interpolated corrected neutron counts (left column) and derived soil moisture (right column) for the three**
**intensive scale surveys during April 2016, June 2016 and March 2017. Blue and red stars indicate the location of the**
**Bishes and Bennetts cosmic-ray neutron sensors.**




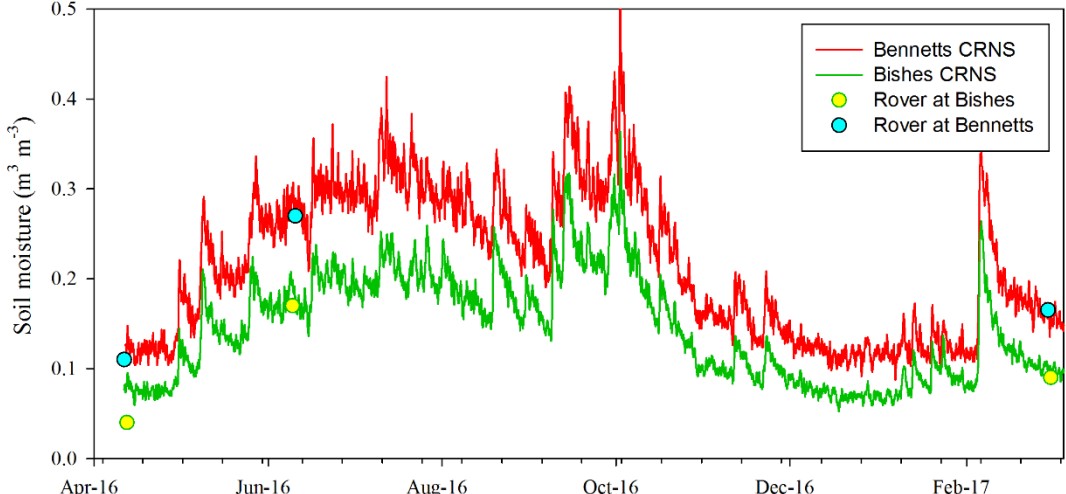


**Figure 8. Comparison of Bennetts and Bishes CRNS soil moisture estimates and corresponding intensive rover survey estimates for the CRNS locations for the three survey dates. Rover survey estimate is from 1 km resolution pixel corresponding to each CRNS location.**


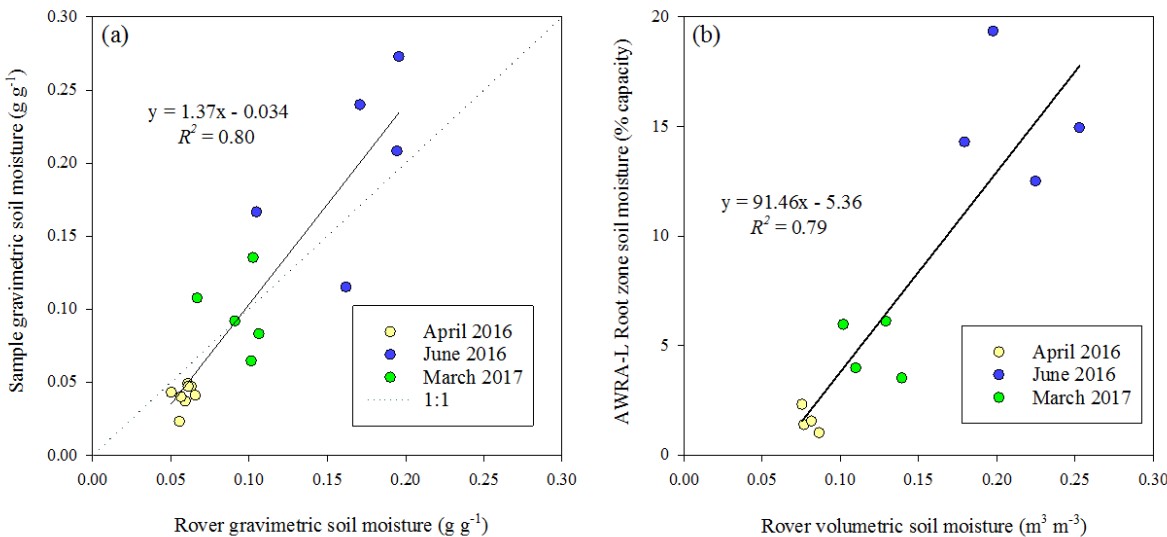


**Figure 9. Intensive rover survey gravimetric soil moisture (1 km resolution) versus point sample gravimetric soil**
**moisture (a) and intensive rover survey soil moisture (up-scaled to 5 km resolution) versus AWRA-L root zone soil**
**moisture (5 km resolution).**



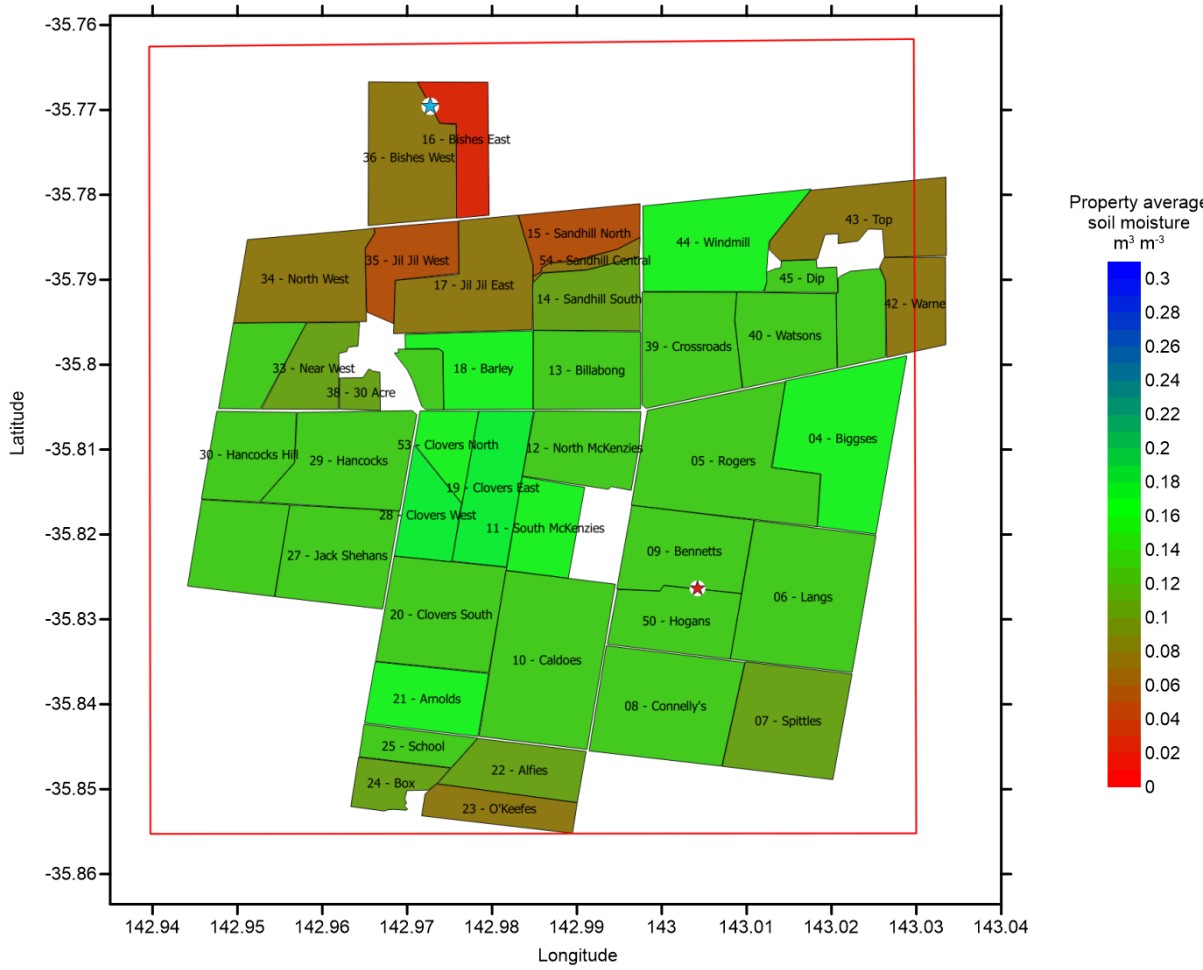


**Figure 10. Location of target properties within the intensive scale survey area (red box) and property average soil moisture content for March 2017. Blue and red stars indicate the location of the Bishes and Bennetts cosmic-ray neutron sensors.**


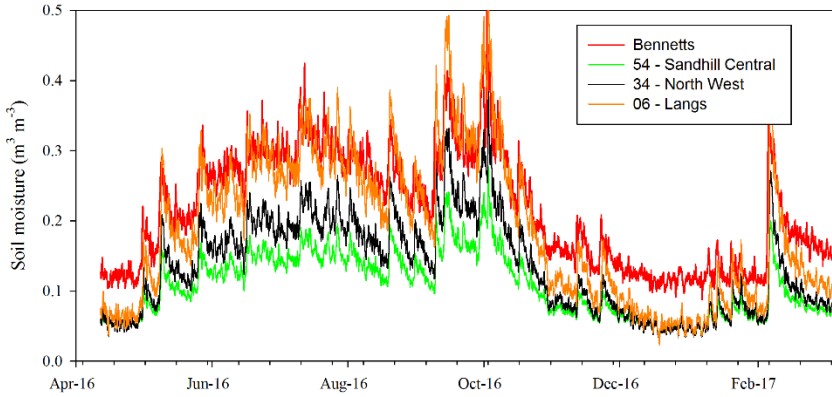


**Figure 11. Time series of average soil moisture for selected properties in the intensive scale survey area and**
**corresponding soil moisture time series from the Bennetts cosmic-ray neutron sensor. Scaling relations ships are**
**provided in Table A1.**

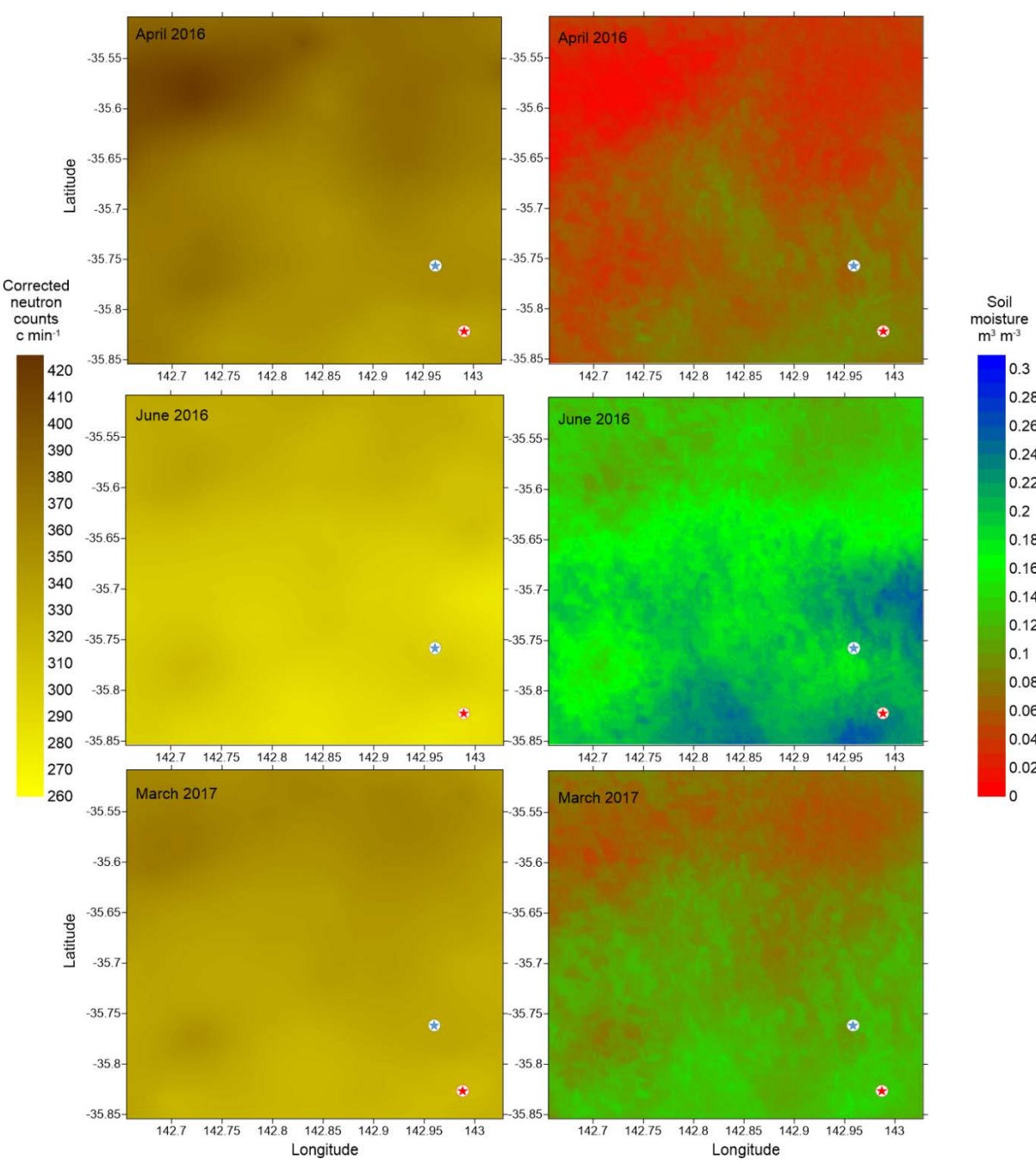

Figure 12. Interpolated corrected neutron counts (left column) and derived soil moisture (right column) for the three broad scale surveys during April 2016, June 2016 and March 2017. Blue and red stars indicate the location of the Bishes and Bennetts cosmic-ray neutron sensors.

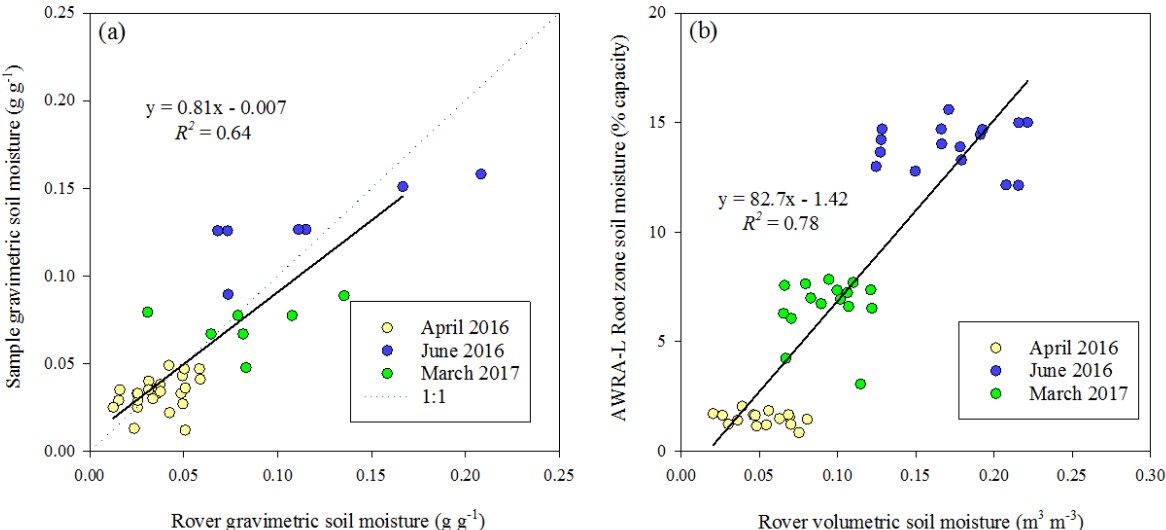


**Figure 13. Broad scale rover survey gravimetric soil moisture (9 km resolution) versus point sample gravimetric soil**
**moisture (a) and broad scale rover survey soil moisture (9 km resolution) versus AWRA-L root zone soil moisture (5**
**km resolution).**


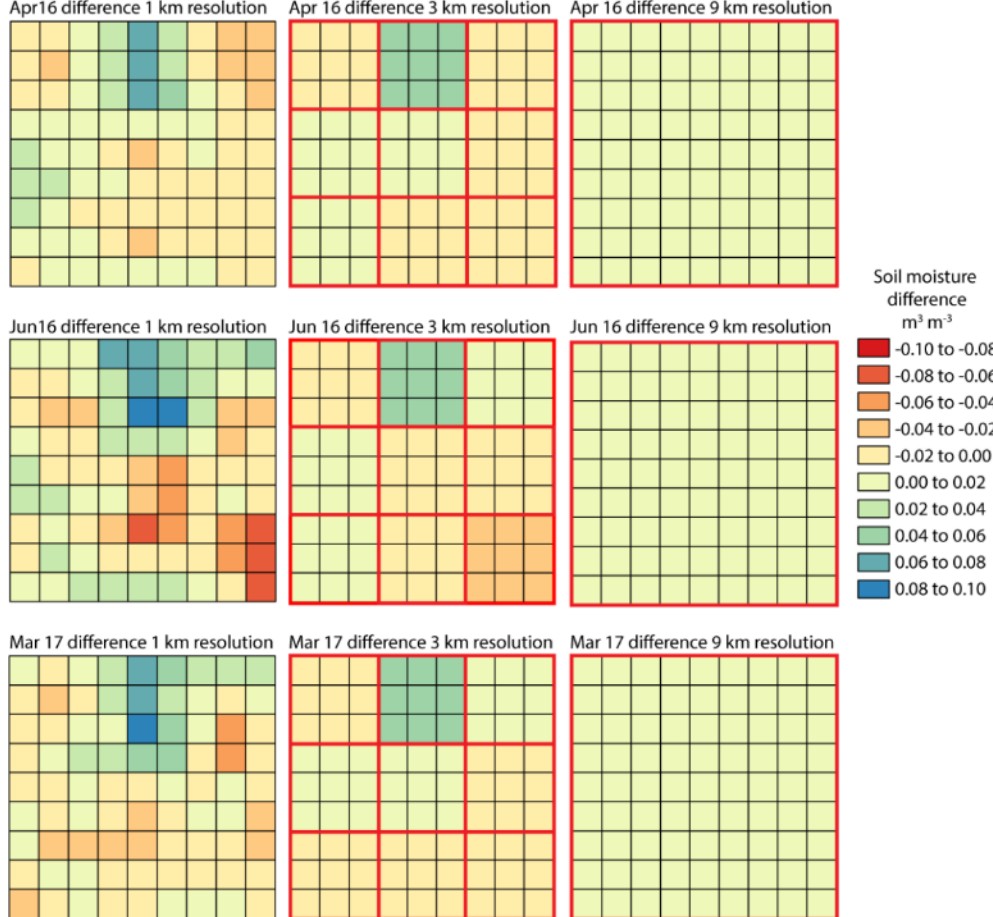

**Figure 14. Difference in soil moisture estimates between the broad and intensive scale surveys for different resolutions on each of the three survey dates. Each cell represents a 1 km x 1 km region within the intensive survey zone.**

**8 Appendix 1**
**Table A1. Supplementary information from regression analysis relating CRNS observations to property average soil**
**moisture content in the intensive scale survey zone.**

| Property | Soil Moisture (m³ m⁻³) | | | Regression modelling results | | |
|---|---|---|---|---|---|---|
| | **Apr-16** | **Jun-16** | **Mar-17** | **Slope** | **Intercept** | $R^2$ |
| *Bennetts CRNS* | *0.124* | *0.277* | *0.157* | | | |
| 54 - Sandhill Central | 0.065 | 0.152 | 0.080 | 0.575 | -0.008 | 0.999 |
| 26 - Whirily | 0.103 | 0.294 | 0.140 | 1.257 | -0.055 | 1.000 |
| 34 - North West | 0.070 | 0.199 | 0.095 | 0.848 | -0.036 | 0.999 |
| 09 - Bennetts | 0.097 | 0.264 | 0.139 | 1.076 | -0.034 | 0.998 |
| 21 - Arnolds | 0.079 | 0.216 | 0.147 | 0.809 | -0.003 | 0.905 |
| 25 - School | 0.082 | 0.222 | 0.136 | 0.858 | -0.013 | 0.968 |
| 17 - Jil Jil East | 0.077 | 0.181 | 0.097 | 0.685 | -0.009 | 0.999 |
| 14 - Sandhill South | 0.074 | 0.202 | 0.104 | 0.828 | -0.027 | 1.000 |
| 24 - Box | 0.079 | 0.223 | 0.118 | 0.922 | -0.032 | 0.997 |
| 29 - Hancocks | 0.086 | 0.210 | 0.139 | 0.749 | 0.006 | 0.947 |
| 13 - Billabong | 0.092 | 0.254 | 0.128 | 1.052 | -0.038 | 1.000 |
| 38 - 30 Acre | 0.081 | 0.187 | 0.106 | 0.688 | -0.003 | 1.000 |
| 18 - Barley | 0.105 | 0.227 | 0.141 | 0.777 | 0.013 | 0.992 |
| 16 - Bishes East | 0.027 | 0.132 | 0.057 | 0.674 | -0.053 | 0.995 |
| 08 - Connelly's | 0.093 | 0.223 | 0.123 | 0.845 | -0.011 | 1.000 |
| 11 - South McKenzies | 0.106 | 0.261 | 0.144 | 1.003 | -0.016 | 0.999 |
| 32 - Far West | 0.063 | 0.192 | 0.124 | 0.765 | -0.016 | 0.919 |
| 36 - Bishes West | 0.043 | 0.166 | 0.091 | 0.754 | -0.040 | 0.962 |
| 40 - Watsons | 0.092 | 0.222 | 0.125 | 0.839 | -0.009 | 0.998 |
| 50 - Hogans | 0.087 | 0.236 | 0.127 | 0.957 | -0.028 | 0.996 |
| 51 - Hennessy's | 0.089 | 0.254 | 0.159 | 1.000 | -0.019 | 0.947 |
| 23 - O'Keefes | 0.062 | 0.187 | 0.099 | 0.793 | -0.031 | 0.992 |
| 22 - Alfies | 0.071 | 0.197 | 0.108 | 0.801 | -0.024 | 0.993 |
| 15 - Sandhill North | 0.045 | 0.122 | 0.063 | 0.504 | -0.017 | 0.999 |
| 35 - Jil Jil West | 0.057 | 0.164 | 0.072 | 0.721 | -0.036 | 0.995 |
| 30 - Hancocks Hill | 0.054 | 0.188 | 0.128 | 0.770 | -0.020 | 0.865 |
| 04 - Biggses | 0.097 | 0.242 | 0.153 | 0.891 | -0.002 | 0.964 |
| 41 - Front | 0.095 | 0.193 | 0.127 | 0.620 | 0.023 | 0.985 |
| 03 - Perns | 0.076 | 0.213 | 0.135 | 0.827 | -0.013 | 0.945 |
| 45 - Dip | 0.095 | 0.213 | 0.135 | 0.734 | 0.011 | 0.982 |
| 06 - Langs | 0.091 | 0.290 | 0.125 | 1.316 | -0.076 | 0.998 |
| 07 - Spittles | 0.094 | 0.275 | 0.119 | 1.216 | -0.063 | 0.993 |
| 05 - Rogers | 0.084 | 0.224 | 0.121 | 0.896 | -0.024 | 0.997 |
| 19 - Clovers East | 0.095 | 0.274 | 0.170 | 1.093 | -0.024 | 0.951 |
| 10 - Caldoes | 0.081 | 0.205 | 0.129 | 0.758 | -0.003 | 0.965 |
| 12 - North McKenzies | 0.089 | 0.269 | 0.140 | 1.149 | -0.048 | 0.995 |
| 27 - Jack Shehans | 0.083 | 0.216 | 0.135 | 0.818 | -0.007 | 0.966 |
| 42 - Warne | 0.066 | 0.189 | 0.089 | 0.807 | -0.035 | 0.999 |
| 44 - Windmill | 0.077 | 0.220 | 0.147 | 0.848 | -0.010 | 0.911 |
| 43 - Top | 0.074 | 0.206 | 0.093 | 0.883 | -0.040 | 0.995 |
| 37 - Barrell | 0.095 | 0.206 | 0.129 | 0.701 | 0.013 | 0.991 |
| 48 - Vernies | 0.082 | 0.200 | 0.103 | 0.781 | -0.017 | 0.999 |
| 20 - Clovers South | 0.086 | 0.221 | 0.139 | 0.830 | -0.006 | 0.963 |
| 33 - Near West | 0.067 | 0.206 | 0.106 | 0.889 | -0.039 | 0.995 |
| 31 - Back Jack Shehans | 0.070 | 0.215 | 0.125 | 0.896 | -0.030 | 0.969 |
| 28 - Clovers West | 0.093 | 0.260 | 0.166 | 1.004 | -0.014 | 0.940 |
| 39 - Crossroads | 0.077 | 0.214 | 0.126 | 0.855 | -0.020 | 0.977 |
| 53 - Clovers North | 0.079 | 0.229 | 0.151 | 0.893 | -0.013 | 0.917 |



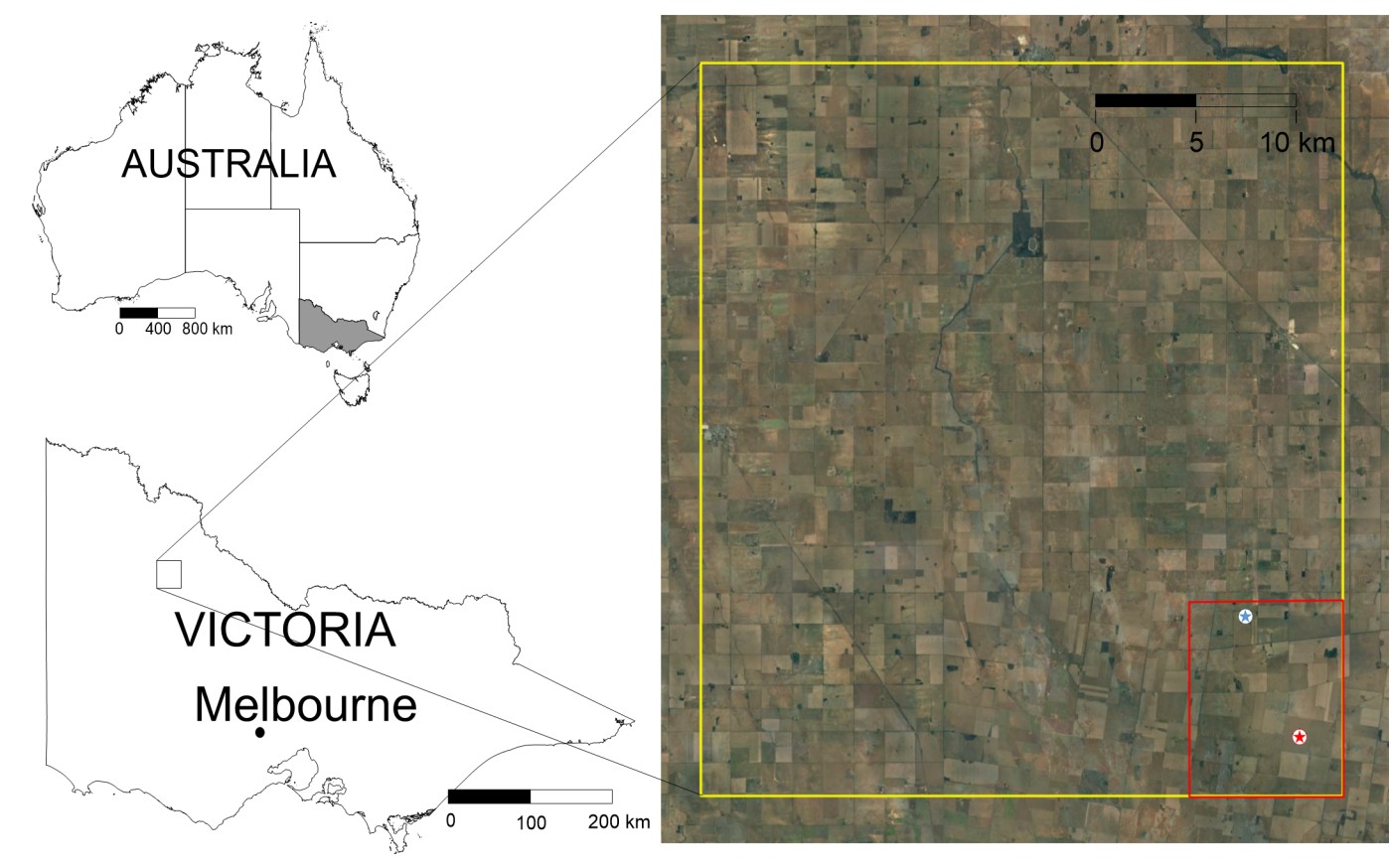

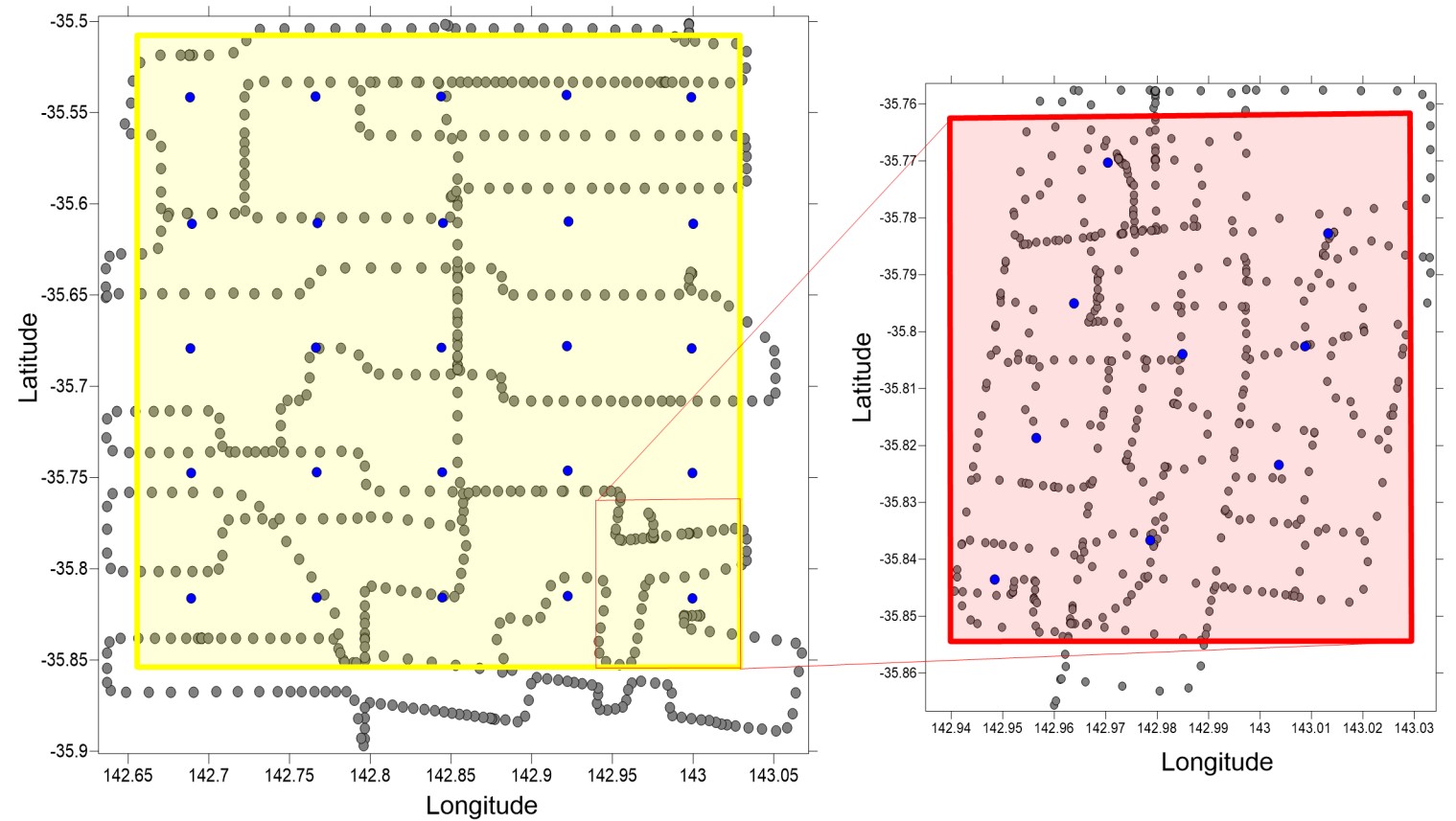

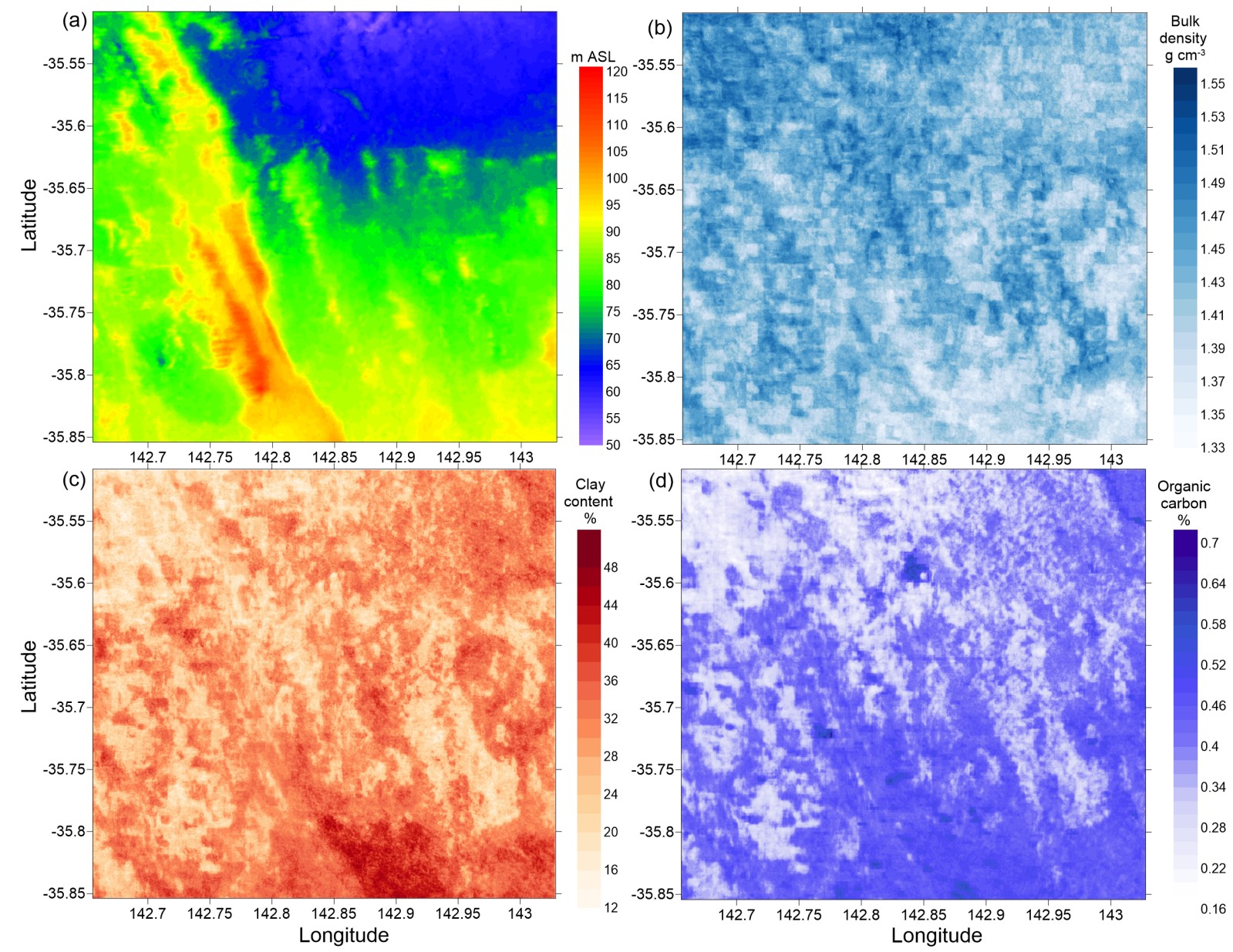

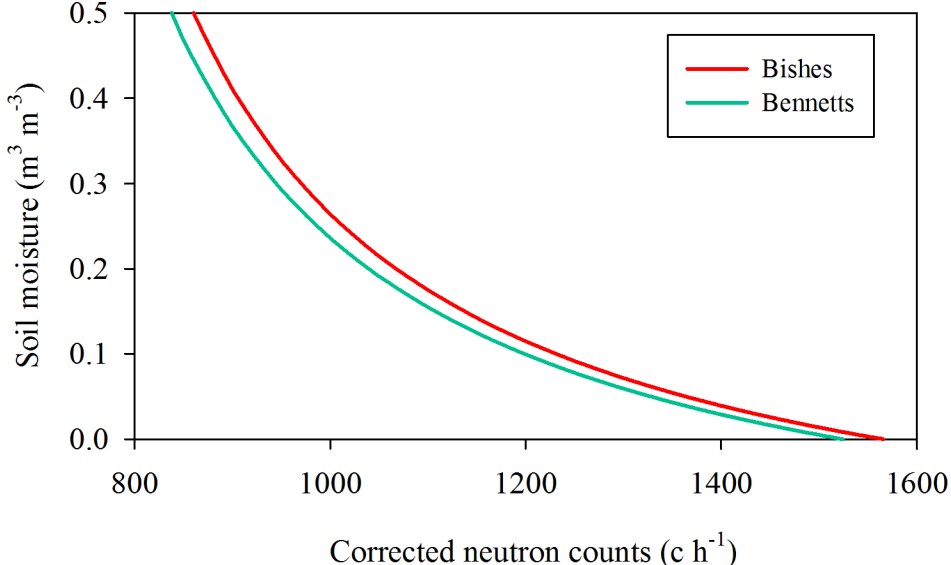

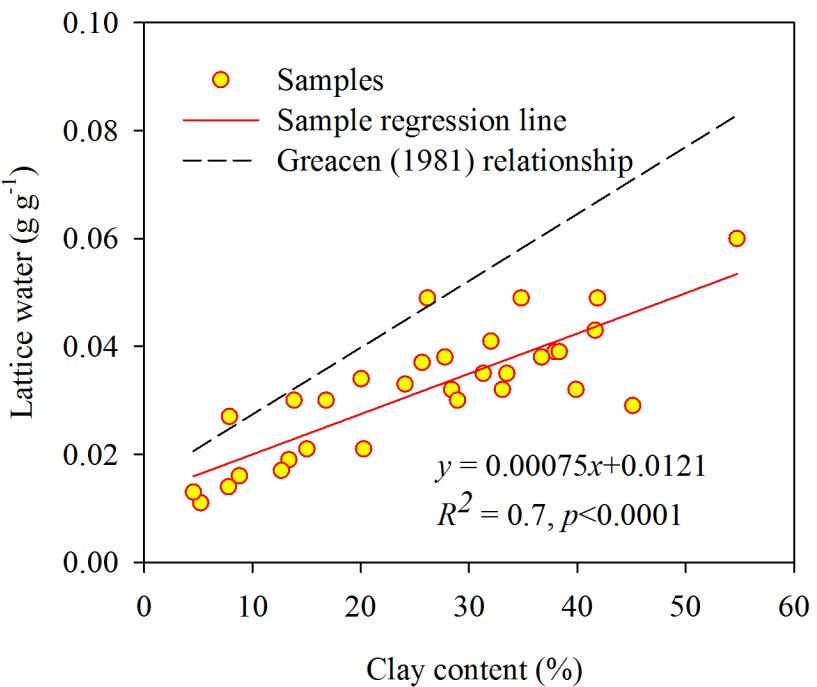

$y = 0.00075x + 0.0121$

$R^2 = 0.7$, $p < 0.0001$

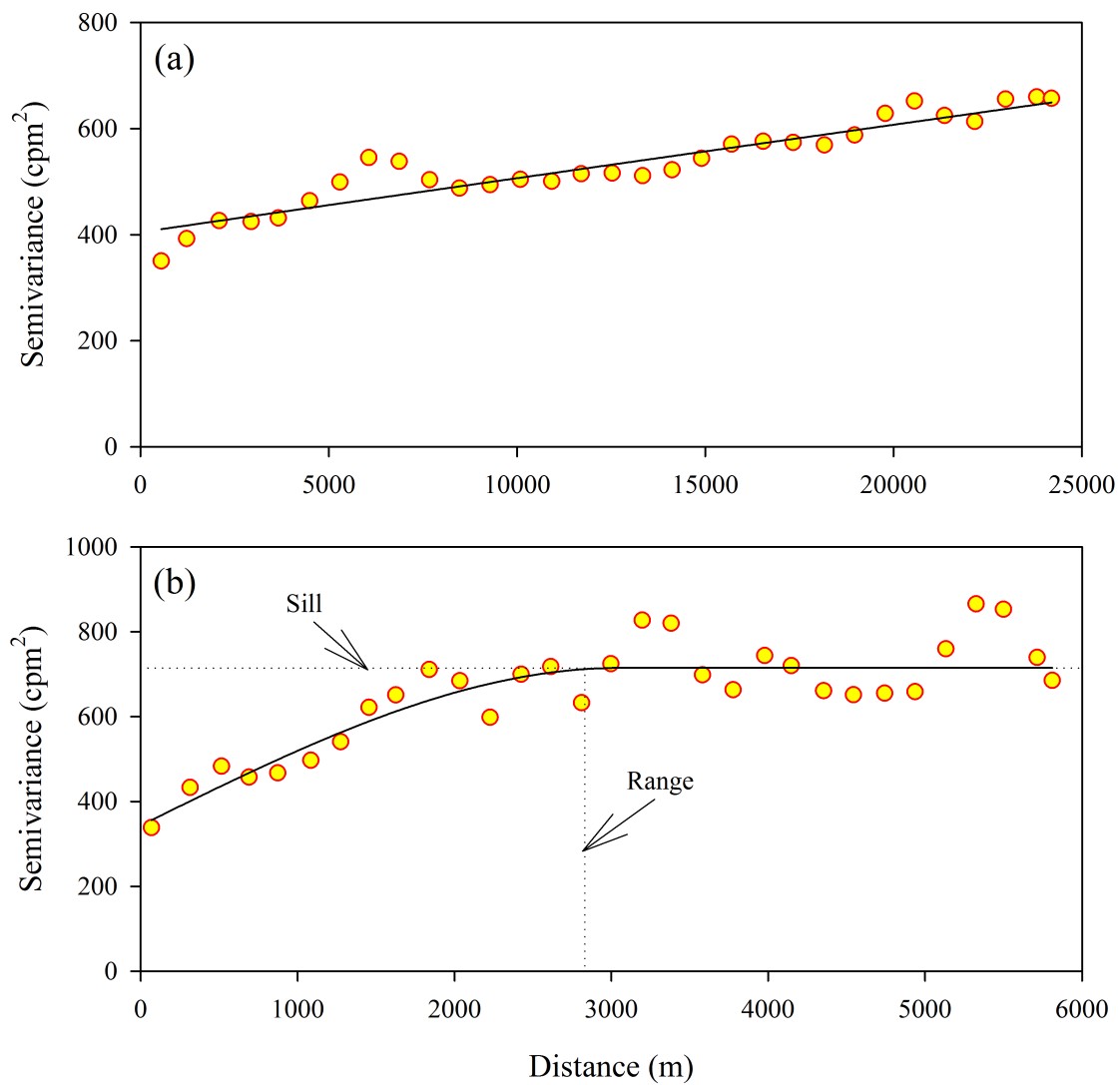

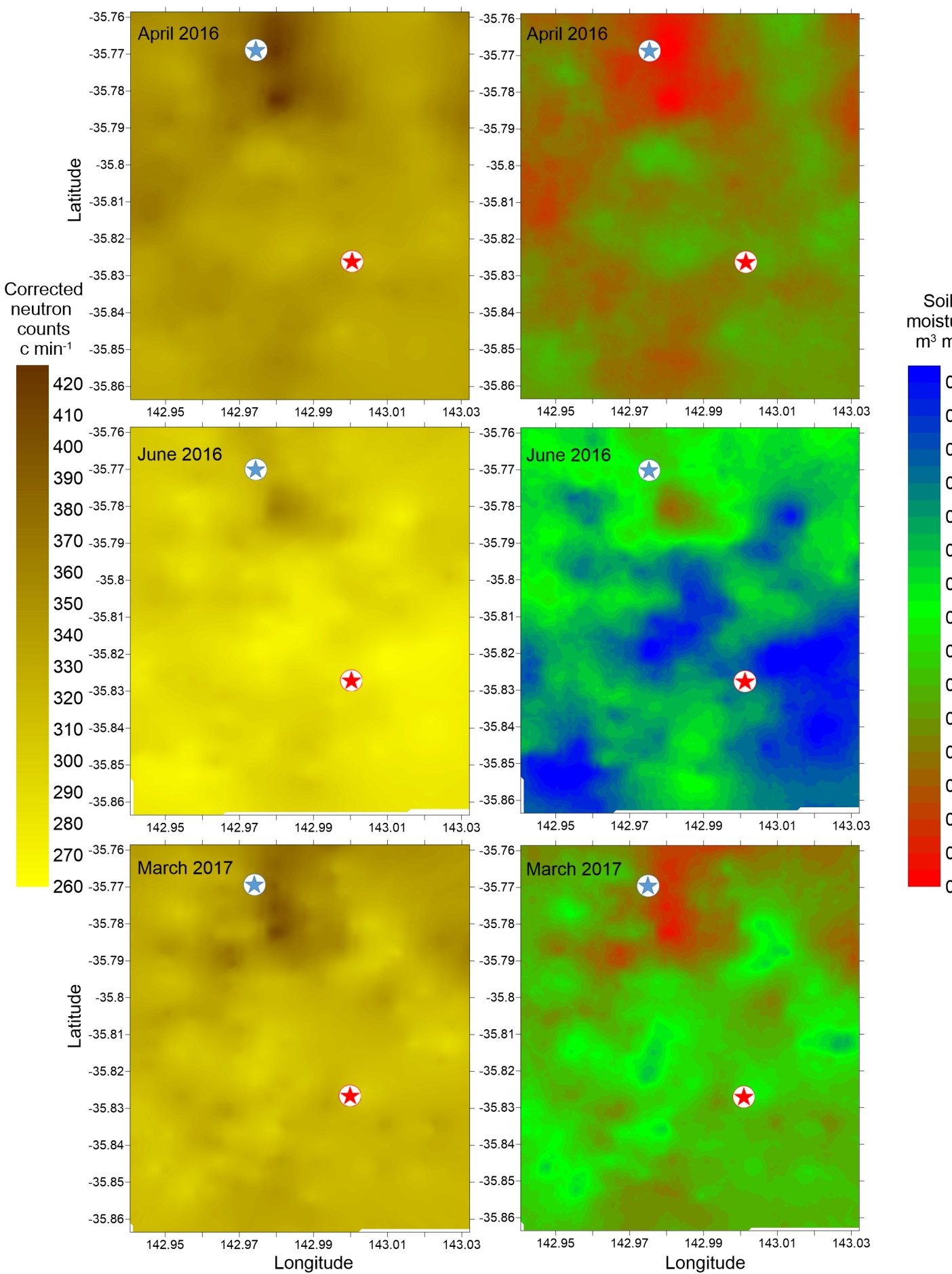

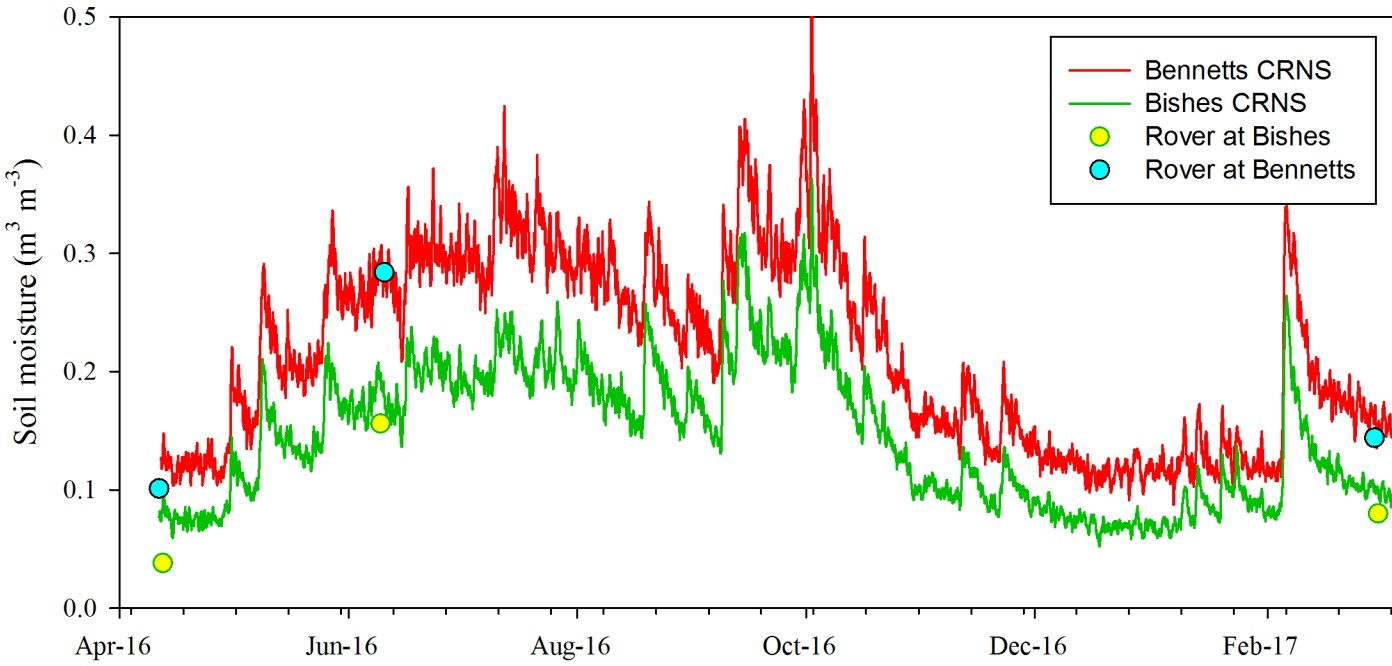

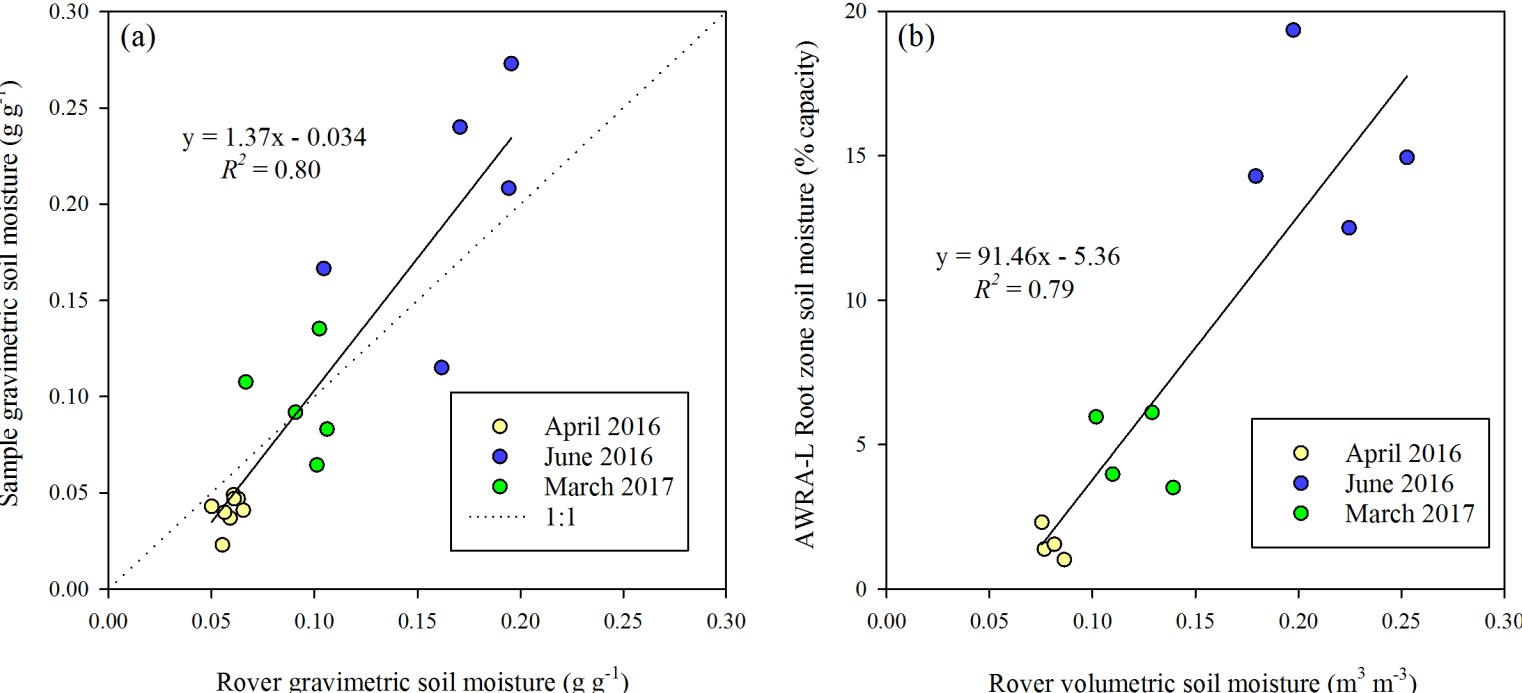

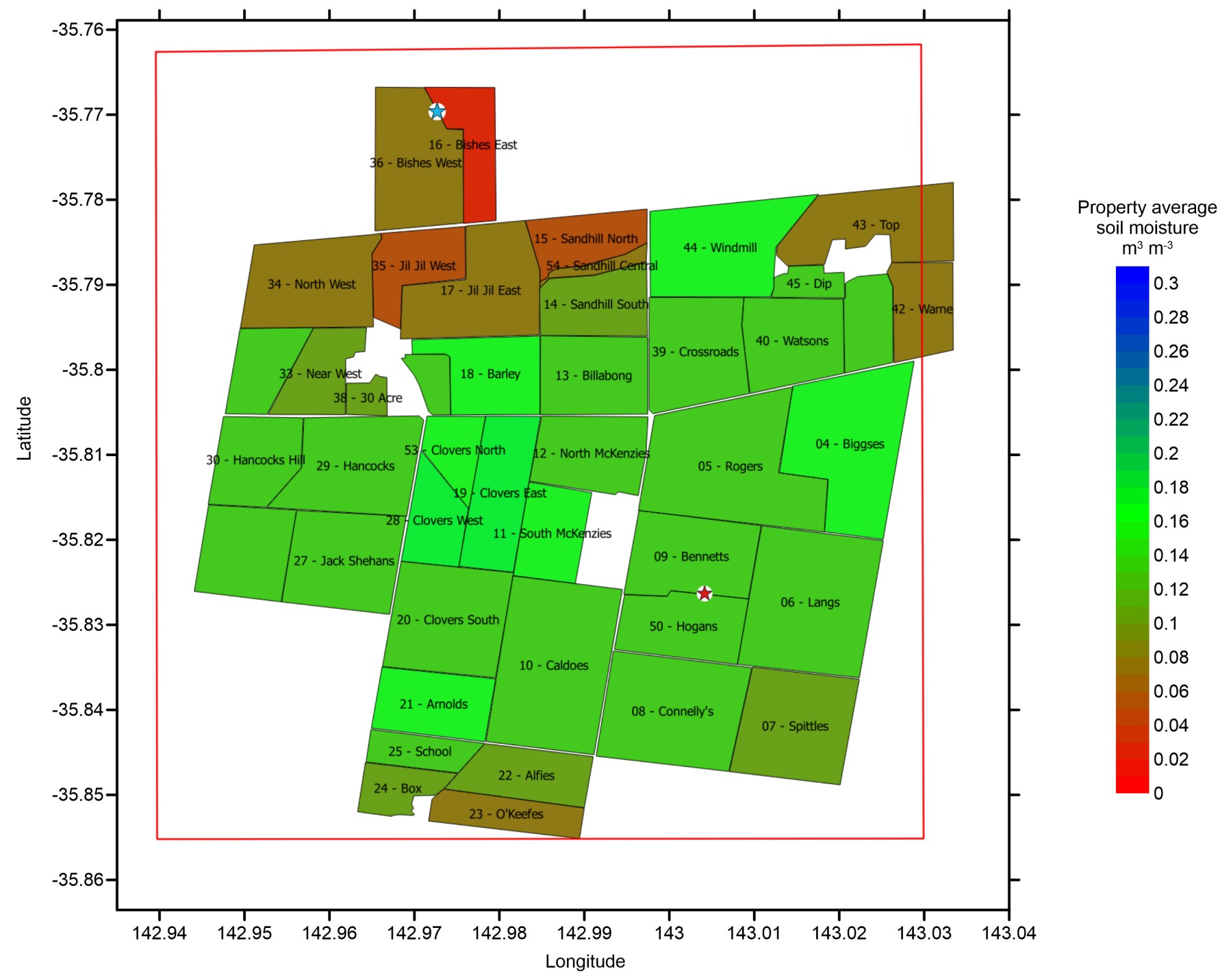

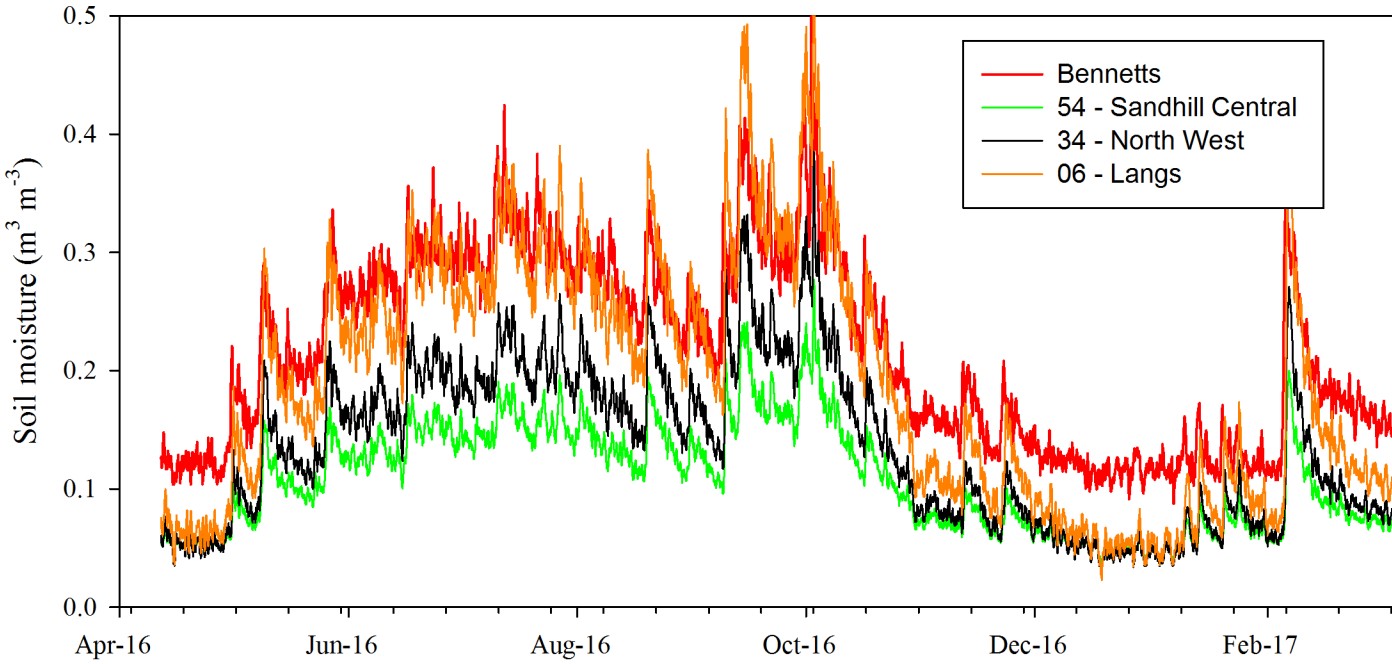

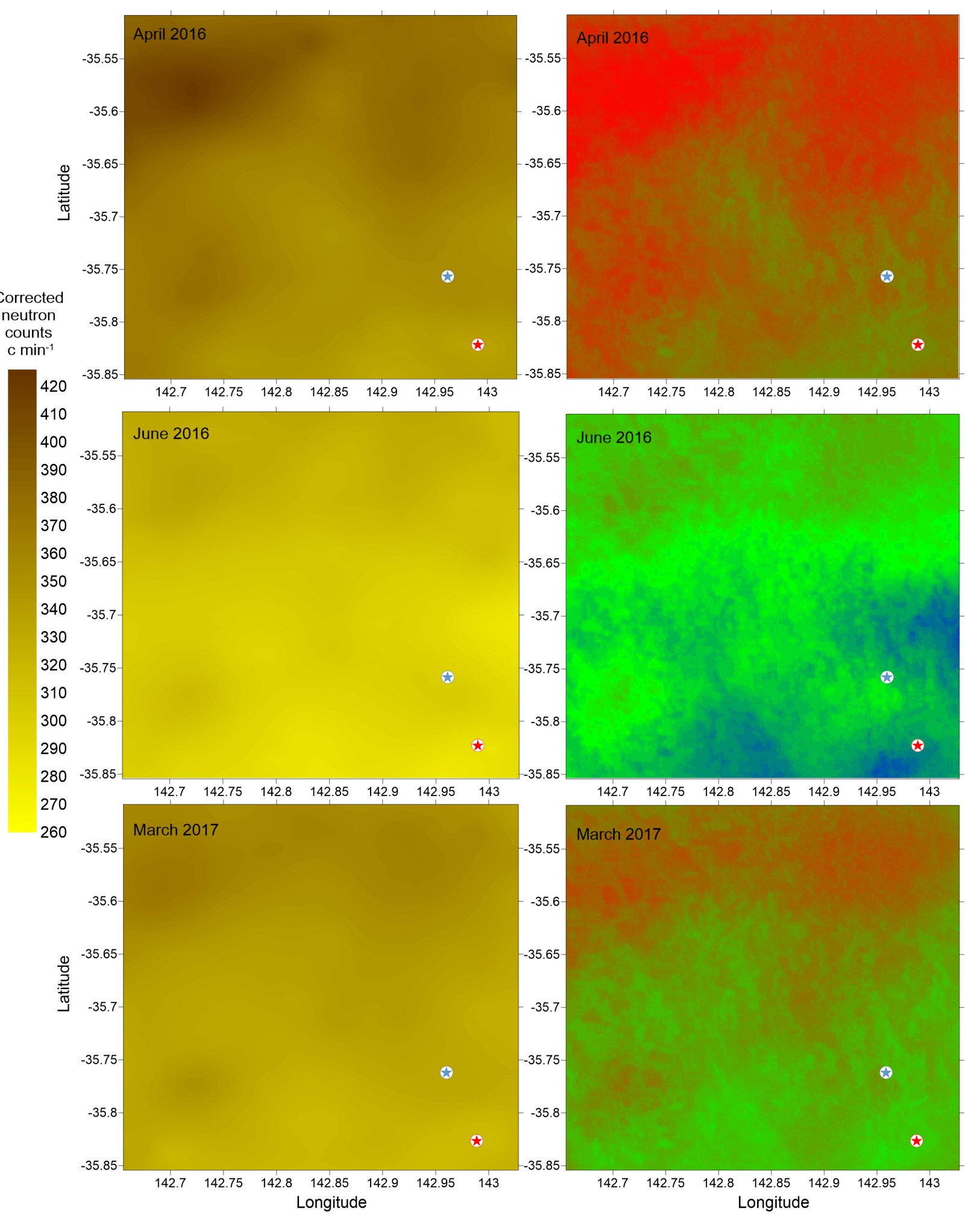

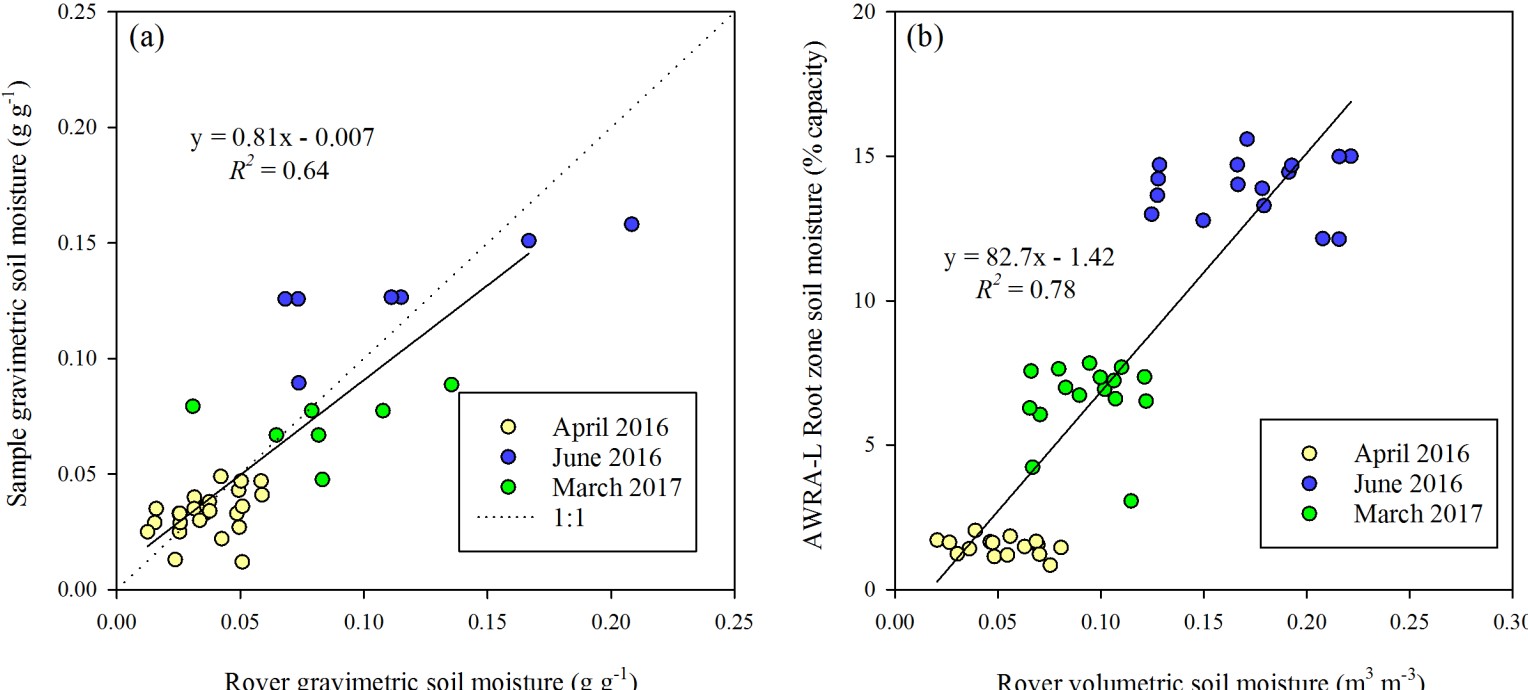

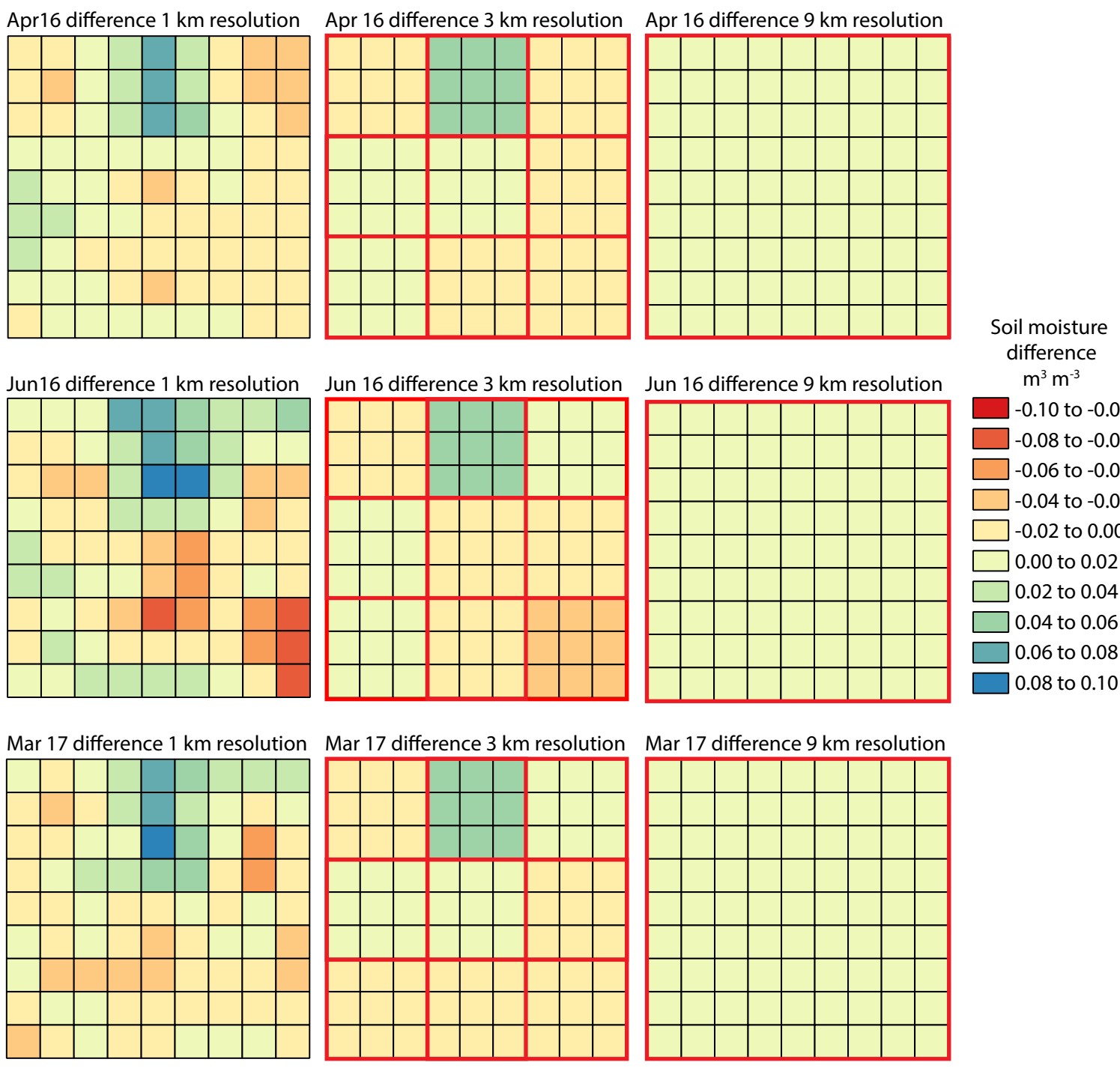

Apr16 difference 1 km resolution

Apr 16 difference 3 km resolution

Apr 16 difference 9 km resolution

Jun16 difference 1 km resolution

Jun 16 difference 3 km resolution

Jun 16 difference 9 km resolution

Mar 17 difference 1 km resolution

Mar 17 difference 3 km resolution

Mar 17 difference 9 km resolution

Soil moisture
difference
$m^3\ m^{-3}$

-0.10 to -0.08
-0.08 to -0.06
-0.06 to -0.04
-0.04 to -0.02
-0.02 to 0.00
0.00 to 0.02
0.02 to 0.04
0.04 to 0.06
0.06 to 0.08
0.08 to 0.10