# Peer review of "cosmic-ray soil moisture sensors"

_Hydrology and Earth System Sciences, 2017_

## Referee Comment (RC1) · Anonymous Referee #1 · 28 Jul 2017

The work presented by MacJannet and others investigates the use of mobile cosmic-ray sensors for estimating soil moisture at a range of scales within a 36 km by 36 km area over an arid region in Australia. There are two regions of interest in the analysis, the 36 km x 36 km region aimed at producing 9km resolution soil moisture maps, and an inner region of 10 km x 10 km aimed at producing 1 km resolution soil moisture estimates. The authors highlight the importance of multi-scale soil moisture estimates for remote sensing validation as well as its use along with high-resolution land surface modeling.

The manuscript is concise and well written with clear steps. The figures are appropriate for the tasks taken and discussed in the manuscript. However, my main issue with this manuscript is its lack of novelty. The use of mobile cosmic-ray sensors (i.e., "rover") for

soil moisture estimates is not new (as pointed out by the authors). The steps taken to convert the neutron counting rates from the rover to the final soil moisture is not new either. The regression analysis done to increase temporal resolution at gridded points within the region has also been done elsewhere. The manuscript reads very much like a technical report in which results are simply reported without much discussion. I don't see a clear scientific question being tackled in this manuscript. Perhaps, the only two pieces of relatively new information I noted were the updated relationship between lattice water and clay content particular applied to their region of interest (in comparison to a previous estimate from Australia) and the impact of number of integration points per area (which is directly related to the speed at which rover surveys are taken) on the quality of the soil moisture maps estimated from coarser to higher resolutions (but refer to my point about this below).

The authors made an important link to remote sensing soil moisture products and land surface modeling, and the manuscript feels a bit incomplete without a proper comparison against additional soil moisture "products". In addition, the authors claimed that the produced maps are "reliable" but how to assess reliability without an independent set of data? I strongly believe an independent set of data and comparison against model and remote sensing could have been an important addition to this manuscript and certainly contributing to its novelty. Unfortunately, I don't see a novel contribution that merits publication in HESS at this stage. My recommendation is for the authors to resubmit the work with a much clearer research question as well as incorporating of other independent soil moisture estimates to verify the impact of the rover soil moisture.

Additional specific comments:

1. Eq. 5: Please, explain what W_lat, W_SOC, and rho_bd are right after the equation is presented. I believe rho_bd is never described properly in the text.

2. Section 2.3: It might be a good idea for the authors to show a picture of the rover system in this section.

3. Section 3.1: I believe Fig 4 is meant to be mentioned in this section (but it is not currently)

4. Section 3.3: I believe Fig 5 is meant to be mentioned in this section (but it is not currently)

5. Section 3.4 and 3.5: The authors assume the reader has good knowledge of spatial statistics and how the fields are ultimately interpolated to produce soil moisture maps. For example, the discussion about "sill" may not be clear to the broad readership of HESS. In fact, what does having or not having a "sill" imply? What does "sill" represent in this case (from a physical soil moisture variability context)? The authors should also highlight the sill parameter in the plots presented in Fig 6.

6. Section 3.5: Ideally, one (including myself) would like to see the soil moisture maps compared against independent measurements. It is expected that the map-derived soil moisture will compare well with the two static sites since the rover was calibrated using the same data. So, the whole approach appears a bit "circular" to me. At the end of this section, the authors make a good point about the importance of these measurements form model testing and remote sensing. I strongly recommend the authors to expand their manuscript to include comparison against remote sensing and land surface model and discuss reasons for similarities and differences.

7. L290-302: There is some potentially interesting analysis here but I also wonder if the results can be strongly influence by the soil properties themselves. In other words, if the authors apply the same comparison between the broad survey and intensive survey using the soil properties (not the estimated soil moisture), would they see a similar behavior? How much of the difference in soil moisture they currently observed is conditioned to the soil properties versus the changes in resolution due to averaging? Also, how can the authors justify comparing measurements, despite being originally taken at different resolution, that essentially come from the same methodology, instrument, and calibration against the same data? This appears a bit weak to me and reinforces my

point about differences due to variation in soil properties.

8. L329-342: Interesting discussion about the road effect. It can definitely influence the results but I'd expect such influence to be more pronounced in humid sites (and not so much at arid sites)??? Also, because the maps (broad and intensive surveys) are derived from the same approach, any road effect may actually be cancelled out when comparing both surveys.

9. Table 1: Please, add a column with footprint-average soil moisture conditions for each case

10. Figure 7: These maps are interesting but they should be evaluated with other points (any points available within the domain) that had not been directly used to calibrate the rover itself. Otherwise, the only information in those maps are potentially the relative differences between wet and dry areas. Similar comment applies to Figs 9, 10, and 11.

11. Figure 8: The results here are expected and my only interpretation here is that the characteristics of soil moisture at 1km resolution (obtained with the rover) are comparable to finer scale from the static sensor (i.e., there may not be large differences between the 200-300m integrated soil moisture compared to the 1km resolution product).

12. Figure 12: For all soil property maps in the domain (W_lat, W_SOC, rho_bd), can the authors reproduce the same plots? In other words, averages at 1km, 3km, and 9km within the overlapped area for broad and intensive surveys. Can the results tell authors what possible controlling factors are associated with the differences between both surveys? I believe this can initially be expanded to something interesting and novel.

---

## Referee Comment (RC2) · Anonymous Referee #2 · 28 Jul 2017

The authors present a nice and straight forward multiscale soil moisture experiment in Australia using the relatively new cosmic-ray neutron rover. While the experiment has been performed in Hawaii, Arizona, Oklahoma, and Nebraska, the study does break some new and interesting ground related to the technology and its application. The authors find an excellent relationship between clay percent and lattice water, which is a critical second order effect on the conversion of neutron counts to soil moisture. In addition, the authors nicely illustrate the challenges and solutions to designing a multi-scale soil moisture experiment. The rover counts and survey speed should be tailored to the scale of soil moisture heterogeneity present and desired scale results of the experiment. Given the need for more and better soil moisture validation datasets for satellite estimates of soil moisture this is an important methodology paper illustrating

the utility of the rover to meet these needs. Lastly the authors present an interesting scaling approach for obtaining point to area averages. This is critical for making soil moisture observations more useful for land management applications which often involve complex areas and are poorly represented by both point sensors and satellite products. The paper is well organized, straight forward, and appropriate for HESS. Attached are a few key points to address.

1. The authors point to area regressions are based on 3 rover surveys only. While I understand the challenge of collecting multi-date information the authors should mention this limitation. In particular future work should perform a leave one out cross validation study in order to properly identify the error of the point to area methodology and temporal stability of soil moisture patterns. I think a description of this need for future work should be discussed more clearly as a limitation of the study. However, I am confident that the cross validation error would be fairly small and not affect the overall conclusions of the paper.

2. Page 2 L 43. The authors should see Andreasen 2016 and 2017 for a better description of the moderated detector energy bins.

Andreasen, M., K. H. Jensen, M. Zreda, D. Desilets, H. Bogena, and M. C. Looms (2016), Modeling cosmic ray neutron field measurements, Water Resources Research, 52(8), 6451-6471. doi:10.1002/2015wr018236.

Andreasen, M., K. H. Jensen, D. Desilets, T. E. Franz, M. Zreda, H. Bogena, and M. C. Looms (In press 2017), Status and perspectives of the cosmic-ray neutron method for soil moisture estimation and other environmental science applications Vadose Zone Journal.

3. Page 8 L 273. The high $R^2$ values are due to the few number of surveys performed. A cross validation experiment would be better suited to address error in future rover work. Authors should mention number of sample points here and in the discussion.
4. Figure 9. The authors should use the same scale as Fig. 7. Odd visual that wetter spots are more red instead of green or blue.

---

## Editor Comment (EC1) · N. Romano (Editor) · 8 Aug 2017

Dear Authors, In the spirit of the discussion step of HESS, to feed this step I suggest you should start providing some preliminary responses to the comments received so far.

---

## Author Comment (AC1) · 9 Aug 2017

The authors present a nice and straight forward multiscale soil moisture experiment in Australia using the relatively new cosmic-ray neutron rover. While the experiment has been performed in Hawaii, Arizona, Oklahoma, and Nebraska, the study does break some new and interesting ground related to the technology and its application. The authors find an excellent relationship between clay percent and lattice water, which is a critical second order effect on the conversion of neutron counts to soil moisture. In addition, the authors nicely illustrate the challenges and solutions to designing a multi-

scale soil moisture experiment. The rover counts and survey speed should be tailored to the scale of soil moisture heterogeneity present and desired scale results of the experiment. Given the need for more and better soil moisture validation datasets for satellite estimates of soil moisture this is an important methodology paper illustrating the utility of the rover to meet these needs. Lastly the authors present an interesting scaling approach for obtaining point to area averages. This is critical for making soil moisture observations more useful for land management applications which often involve complex areas and are poorly represented by both point sensors and satellite products. The paper is well organized, straight forward, and appropriate for HESS.

Attached are a few key points to address.

1. The authors point to area regressions are based on 3 rover surveys only. While I understand the challenge of collecting multi-date information the authors should mention this limitation. In particular future work should perform a leave one out cross validation study in order to properly identify the error of the point to area methodology and temporal stability of soil moisture patterns. I think a description of this need for future work should be discussed more clearly as a limitation of the study. However, I am confident that the cross validation error would be fairly small and not affect the overall conclusions of the paper.

RESPONSE: This limitation is now clearly identified and, as suggested by the reviewer a recommendation for replication and further testing has been added to the discussion. We agree that the cross validation will likely show only small errors but this discussion is included anyway.

2. Page 2 L 43. The authors should see Andreasen 2016 and 2017 for a better description of the moderated detector energy bins.

Andreasen, M., K. H. Jensen, M. Zreda, D. Desilets, H. Bogena, and M. C. Looms (2016), Modeling cosmic ray neutron field measurements, Water Resources Research, 52(8), 6451-6471. doi:10.1002/2015wr018236.

[Figure]

Andreasen, M., K. H. Jensen, D. Desilets, T. E. Franz, M. Zreda, H. Bogena, and M. C. Looms (In press 2017), Status and perspectives of the cosmic-ray neutron method for soil moisture estimation and other environmental science applications Vadose Zone Journal.

RESPONSE: A better description of the energy bins detected by the CRNS has now been included and the relevant references have been included as suggested

3. Page 8 L 273. The high RĔȨ2 values are due to the few number of surveys performed. A cross validation experiment would be better suited to address error in future rover work. Authors should mention number of sample points here and in the discussion.

RESPONSE: New text highlighting the limitations of three data points has now been added to the results and discussion section to make this clear to the reader.

4. Figure 9. The authors should use the same scale as Fig. 7. Odd visual that wetter spots are more red instead

RESPONSE: Figure 9 has been revised and now uses the same colour scale as Fig 7 for consistency. The wet blue colour from fig 7 scale does not appear as the soils are too dry.

A REVISED VERSION OF THE DOCUMENT WILL BE UPLOADED ONCE RESPONSES TO REVIEWER 1 ARE COMPLETE

---

## Author Comment (AC2) · 11 Aug 2017

RESPONSE TO REVIEWER 1 Responses included in text below...
The work presented by MacJannet and others investigates the use of mobile cosmic ray sensors for estimating soil moisture at a range of scales within a 36 km by 36 km area over an arid region in Australia. There are two regions of interest in the analysis,

the 36 km x 36 km region aimed at producing 9km resolution soil moisture maps, and an inner region of 10 km x 10 km aimed at producing 1 km resolution soil moisture estimates. The authors highlight the importance of multi-scale soil moisture estimates for remote sensing validation as well as its use along with high-resolution land surface modeling.

The manuscript is concise and well written with clear steps. The figures are appropriate for the tasks taken and discussed in the manuscript. However, my main issue with this manuscript is its lack of novelty. The use of mobile cosmic-ray sensors (i.e., "rover") for soil moisture estimates is not new (as pointed out by the authors). The steps taken to convert the neutron counting rates from the rover to the final soil moisture is not new either. The regression analysis done to increase temporal resolution at gridded points within the region has also been done elsewhere. The manuscript reads very much like a technical report in which results are simply reported without much discussion. I don't see a clear scientific question being tackled in this manuscript. Perhaps, the only two pieces of relatively new information I noted were the updated relationship between lattice water and clay content particular applied to their region of interest (in comparison to a previous estimate from Australia) and the impact of number of integration points per area (which is directly related to the speed at which rover surveys are taken) on the quality of the soil moisture maps estimated from coarser to higher resolutions (but refer to my point about this below).

The authors made an important link to remote sensing soil moisture products and land surface modeling, and the manuscript feels a bit incomplete without a proper comparison against additional soil moisture "products". In addition, the authors claimed that the produced maps are "reliable" but how to assess reliability without an independent set of data? I strongly believe an independent set of data and comparison against model and remote sensing could have been an important addition to this manuscript and certainly contributing to its novelty. Unfortunately, I don't see a novel contribution that merits publication in HESS at this stage. My recommendation is for the authors

to resubmit the work with a much clearer research question as well as incorporating of other independent soil moisture estimates to verify the impact of the rover soil moisture.

RESPONSE: We are currently preparing a manuscript comparing the rover surveys to sentinel satellite soil moisture retrievals and an Australian wide high resolution soil moisture modelling product. To include this comparison in this paper would make an extremely large paper. The paper is about establishing correct experimental design, developing new approaches to provision of spatial soil properties, nesting surveys to test resolution and survey design and exploring temporal stability in soil moisture across scales. We prefer to set out the approaches and novel findings here. The reliability in measurements comes from the calibration of the static and rover sensors and the fact that we can replicate the measurements across time. Unfortunately, there are no other 'scientific standard' measurements across this data poor location which is the whole purpose of proposing the rover surveys – the results here will be the basis of model/satellite validation and, hopefully, much better soil moisture information into the future but the first step is getting the rover experimental design, calibration and processing sorted which is what we have done in this paper. The testing across scales (fig 12) shows this.

We strongly believe there are enough contributions to make this a stand-alone paper. In response to the comment that this paper has no novel contribution we have to respectfully disagree. The novel contributions of this paper are:

1. We have develop a clay to lattice water relationship which is very strong. This growing data base of lattice water to clay relationships has also enabled us to produce a new lattice water product using the Australian Soil and Landscape Grid. This has potential application across Australia and internationally.

2. Our rover study is the first to use a digital soil mapping product to account for the spatial variation in soil properties across the survey area. This facilitate an easier set of data processing procedures and minimised assumptions that are made in other rover

studies. This approach will be key to stream-lining the processing of spatial rover data in future surveys. This is a new approach to convert the neutron counting rates from the rover to the final soil moisture which can be applied to other gridded soil property databases.

3. We are the first study to use a nested high resolution survey within a larger broad scale survey. This approach has enabled us to test our experimental design in particular our selected driving speed and desired product resolution. This comparison has highlighted the need to design surveys fit for purpose and shows that different kriging models are required for different scale surveys as they are sensitive to different spatial information.

4. We have further demonstrated that N0 for static probes is strongly controlled by biomass. Our two static sites with different soil type and moisture have essentially the same N0 as the respective footprints are essentially biomass free. This is very useful information for rover surveys in this region and points to a standard N0 if biomass is accounted for in calibration and spatial variation in incoming neutron intensity can be correctly accounted for.

5. We have provided evidence for temporal stability in soil moisture in this dry land setting. We demonstrate this at the property scale (most relevant to farm managers) and this has great relevance to local land holders who can relate their property to neighbouring sensors, and scientists who can use point-to-area scaling to fill the gaps between rover surveys for comparison to other soil moisture products. These points will all be described in our modified discussion section.

Additional specific comments:

1. Eq. 5: Please, explain what W_lat, W_SOC, and rho_bd are right after the equation is presented. I believe rho_bd is never described properly in the text.

RESPONSE: Will be fixed as suggested and rho_bd description will be added

2. Section 2.3: It might be a good idea for the authors to show a picture of the rover system in this section.

RESPONSE: We have pictures on our cosmoz website (http://cosmoz.csiro.au/about-cosmoz/ ) so we will add a link to these rather than making the manuscript any larger

3. Section 3.1: I believe Fig 4 is meant to be mentioned in this section (but it is not currently)

RESPONSE: Fixed

4. Section 3.3: I believe Fig 5 is meant to be mentioned in this section (but it is not currently)

RESPONSE: Fixed

5. Section 3.4 and 3.5: The authors assume the reader has good knowledge of spatial statistics and how the fields are ultimately interpolated to produce soil moisture maps. For example, the discussion about "sill" may not be clear to the broad readership of HESS. In fact, what does having or not having a "sill" imply? What does "sill" represent in this case (from a physical soil moisture variability context)? The authors should also highlight the sill parameter in the plots presented in Fig 6.

RESPONSE: The concept of the sill and what it means for spatial statistics will be added to section 3.4. The meaning of the sill and range in the context of spatial interpolation will be added. The sill and range will be labelled in fig 6 to aid interpretation as suggested by the reviewer

6. Section 3.5: Ideally, one (including myself) would like to see the soil moisture maps compared against independent measurements. It is expected that the map-derived soil moisture will compare well with the two static sites since the rover was calibrated using the same data. So, the whole approach appears a bit "circular" to me. At the end of this section, the authors make a good point about the importance of these measurements form model testing and remote sensing. I strongly recommend the authors to expand

their manuscript to include comparison against remote sensing and land surface mode and discuss reasons for similarities and differences.

RESPONSE: Unfortunately there are no other 'scientific standard' measurements across this data poor location so this comparison is not possible. In addition many points would be required due the very high variability in soil moisture often exhibited at point measurement scale and our desire to compare to a large scale rover product. We acknowledge that there may appear to be some circularity but we are not comparing the non-moving rover to the static sensors – this is the final interpolated soil moisture product using the soil grid properties and conventional kriging of neutron counts. We are testing the whole calculation procedure and underlying data (i.e. soil properties) here. We are comparing to static sensor which uses locally measured soil properties. The results here would only be expected to be this good if the spatial interpolation models used were accurate. Comparison to remote sensing and model estimates is in preparation – too much to cover in this paper as well.

7. L290-302: There is some potentially interesting analysis here but I also wonder if the results can be strongly influence by the soil properties themselves. In other words, if the authors apply the same comparison between the broad survey and intensive survey using the soil properties (not the estimated soil moisture), would they see a similar behavior? How much of the difference in soil moisture they currently observed is conditioned to the soil properties versus the changes in resolution due to averaging? Also, how can the authors justify comparing measurements, despite being originally taken at different resolution, that essentially come from the same methodology, instrument, and calibration against the same data? This appears a bit weak to me and reinforces my point about differences due to variation in soil properties.

RESPONSE: We cannot produce the same plots of difference (as with soil moisture, i.e. Fig12) as the same underlying soil property data is used from the Australian soil and Landscape grid for both surveys and its resolution is 90 m. The differences between surveys observed in Fig 12 are purely those related to differences in neutron counts in

both surveys which were observed at very different speed. The broad survey is moving so fast the small scale detail is smoothed out hence the difference at the 1 km and 3km scale but none at 9km scale. We justify comparing the measurements as we are demonstrating the importance of selecting the appropriate drive speed depending on the final product resolution required. We show that the speeds used for the broad scale survey are not suitable for soil moisture estimates at resolutions of 1 km and 3 km. We only get agreement between the two products when the resolution is set at 9km – i.e. the design speed has been successfully set.

8. L329-342: Interesting discussion about the road effect. It can definitely influence the results but I'd expect such influence to be more pronounced in humid sites (and not so much at arid sites)??? Also, because the maps (broad and intensive surveys) are derived from the same approach, any road effect may actually be cancelled out when comparing both surveys.

RESPONSE: The issues of road influences is definitely an interesting one and is something future surveys should take into account. I am aware of some researchers who are working on a solution to this issue (not published yet) and as you say this will be particularly useful in wetter/more humid areas. To push this fact further more text will be added to highlight that the dry road will be over represented in the measured neutron intensity as the sensitivity to hydrogen of neutron intensity is greater at the dry end.

9. Table 1: Please, add a column with footprint-average soil moisture conditions for each case

RESPONSE: This information will be added to Table 1

10. Figure 7: These maps are interesting but they should be evaluated with other points (any points available within the domain) that had not been directly used to calibrate the rover itself. Otherwise, the only information in those maps are potentially the relative differences between wet and dry areas. Similar comment applies to Figs 9, 10, and 11.

RESPONSE: There are unfortunately no other completely independent points in the domain which can be used for evaluation of soil moisture estimates. The point of the figure is to show the relative differences in the domain and to show the ability of the rover to observe these. We discuss how these patterns relate to soil properties which of course reflects differences in soil moisture (i.e. sand v clay).

11. Figure 8: The results here are expected and my only interpretation here is that the characteristics of soil moisture at 1km resolution (obtained with the rover) are comparable to finer scale from the static sensor (i.e., there may not be large differences between the 200-300m integrated soil moisture compared to the 1km resolution product).

RESPONSE: The results here would only be expected to be this good if the spatial interpolation models used were accurate. We are not comparing the non-moving rover to the static sensors – this is the final interpolated soil moisture product using the soil grid properties and conventional kriging of neutron counts. We are comparing to static sensor which uses locally measured soil properties.

12. Figure 12: For all soil property maps in the domain (W_lat, W_SOC, rho_bd), can the authors reproduce the same plots? In other words, averages at 1km, 3km, and 9km within the overlapped area for broad and intensive surveys. Can the results tell authors what possible controlling factors are associated with the differences between both surveys? I believe this can initially be expanded to something interesting and novel.

RESPONSE: As for comment 7 above - We cannot produce the same plots of difference (as with soil moisture, i.e. Fig12) as the same underlying soil property data is used from the Australian soil and Landscape grid for both surveys and its resolution is 90 m. The differences observed in Fig 12 are purely those related to differences in neutron counts in both surveys which were observed at very different speed. The broad survey is moving so fast the small scale detail is smoothed out hence the difference at the 1 km and 3km scale but none at 9km scale.

---

## Referee Comment (RC3) · Anonymous Referee #3 · 13 Aug 2017

Review of the paper:

**"Multiscale soil moisture estimates using static and roving cosmic-ray soil moisture sensors"**

**by: David McJannet**

**GENERAL COMMENTS**

   The paper describes a research project aimed at producing soil moisture estimates at a range of scales that are commensurate with model and satellite retrievals. The study involved static cosmic ray neutron sensors and rover surveys across both broad (36 km at 9 km resolution) and intensive (10 x 10 km at 1 km resolution) scales in a cropping district in the Mallee region of Victoria, Australia.

Given the ever increasing lack of ground measurements, having medium-to-high resolution observations of soil moisture against which validating satellite soil moisture products is extremely important. With the advent of Sentinel 1 satellite sensor we will have soon soil moisture estimates at 1 km of resolution or even lower. Hence, studies involving any technique for retrieving or expand the availability of these information are very welcome in literature. For this reason, I think the topic is of interest for the journal readership and worth the consideration for the publishing in HESS journal. The paper is also well written and structured and concise at point.

My main recommendation for the authors is to put more effort to underline the real merit of the paper by trying to underline the differences with respect to previous studies and add material that makes the study more close to a scientific paper than a technical report. Indeed, I struggled a bit to grasp the novelty and potentiality of the study – "The paper describes a research project" as written by the authors in the abstract – and this does not do justice to the merit of the study. My suggestion is to provide a comparison of the rover estimates with a model or other types of observations (like the gravimetric measurements the authors have collected) demonstrating the reliability of the rover estimates in terms of reproducing spatial pattern of soil moisture which can be extremely useful for validating high-resolution satellite soil moisture products.

I also have other comments the authors can be considered to improve the manuscript. I report below my comments in order of appearance indicating also their relevance.

| PAGE | LINES or Section | RELEVANCE | COMMENT |
|---|---|---|---|
| 3 | 102 | MINOR | Define fp here. Cosmic-ray neutron intensity, fp, is part…. |
| 5 | 155 | MINOR | 18 time.. faster? |
| 7 | Section 3.2 and 3.3. | MODERATE | Figures 4 and 5 seem not cited in the text. |

| 7-8 | Section 3.5 Intensive scale rover survey | MAJOR | I think it is too much optimistic to say that the agreement is excellent based on only on two points and three times. Why not comparing spatially with model estimates? |
|---|---|---|---|
| 8 | 264-277 | MODERATE/ MAJOR | Provide more details about the point-area regression analysis. It is not completely clear from the text. |
| 16 | Figure 1 | MINOR | Provide scale of the figure and indication of the size of the box. |

Based on the comments above I recommend the publication after **MODERATE/MAJOR REVISIONS**.

---

## Author Comment (AC3) · 15 Aug 2017

The lead author has just been reminded about some opportunistic grab samples collected during surveys for analysis. These will be processed and used as point samples for comparison to rover data. We will also now compare the broad scale survey to the soil moisture estimates as produced by the Australian Bureau of Meteorology water balance model on a 5km grid. Stay tuned for new figures and comparisons

---

## Author Comment (AC4) · 18 Aug 2017

Review of the paper: "Multiscale soil moisture estimates using static and roving cosmic-ray soil moisture sensors" by: David McJannet GENERAL COMMENTS The paper describes a research project aimed at producing soil moisture estimates at a range of scales that are commensurate with model and satellite retrievals. The study involved static cosmic ray neutron sensors and rover surveys across both broad (36 km at 9 km resolution) and intensive (10 x 10 km at 1 km resolution) scales in a cropping district in the Mallee region of Victoria, Australia. Given the ever increasing lack of ground measurements, having medium-to-high resolution observations of soil moisture against which validating satellite soil moisture products is extremely important. With the advent of Sentinel 1 satellite sensor we will have soon soil moisture estimates at 1 km of reso-

lution or even lower. Hence, studies involving any technique for retrieving or expand the availability of these information are very welcome in literature. For this reason, I think the topic is of interest for the journal readership and worth the consideration for the publishing in HESS journal. The paper is also well written and structured and concise at point. My main recommendation for the authors is to put more effort to underline the real merit of the paper by trying to underline the differences with respect to previous studies and add material that makes the study more close to a scientific paper than a technical report. Indeed, I struggled a bit to grasp the novelty and potentiality of the study – "The paper describes a research project" as written by the authors in the abstract – and this does not do justice to the merit of the study. My suggestion is to provide a comparison of the rover estimates with a model or other types of observations (like the gravimetric measurements the authors have collected) demonstrating the reliability of the rover estimates in terms of reproducing spatial pattern of soil moisture which can be extremely useful for validating high-resolution satellite soil moisture products.

RESPONSE: In response to the reviewer comments we have now made major changes and compared distributed gravimetric point samples from each survey to rover results for both the intensive and broad scale products. We have also compared rover survey results at intensive and broad scale against distributed point samples and 5 km resolution water balance model estimates of soil moisture. For this analysis we have used the recently operationalised Australian Bureau of Meteorology water balance model estimates of root zone soil moisture. We have added two new figures to this reposnse to show this comparison (see end of this submission)

In addition to this we also believe that the components of this paper that make it novel include; 1) our newly developed clay to lattice water relationship which we apply to nationally available soil property grid for Australia, 2) use of digital soil mapping products to account for the spatial variation in soil properties across the survey area and facilitate data processing, 3) presenting results of a nested high resolution survey within

a larger broad scale survey which enabled us to test our experimental/driving speed design, 4) providing further evidence that N0 for static probes is strongly controlled by biomass, and 5) demonstrating temporal stability in soil moisture in this dry land setting. Significant rewording and new text and figures has been added throughout.

I also have other comments the authors can be considered to improve the manuscript. I report below my comments in order of appearance indicating also their relevance. (COMMENTS REMOVED FROM TABLE FOR RESPONSE): Page3, Line 102, Minor: Define fp here. Cosmic-ray neutron intensity, fp, is part. . .. RESPONSE: Fixed

Page 5, Line 155, Minor: 18 time..faster? RESPONSE: Reworded to "The rover has counting rates approxmately18 times greater than that of a standard static sensor under the same condition, thus, allowing for measurements to be made at one minute intervals."

Page 7, Section 3.2 and 3.3, Moderate Figures 4 and 5 not cited in text RESPONSE: Fixed

Page 7, Section 3.5 Intensive scale rover survey, Moderate, I think it is too much optimistic to say that the agreement is excellent based on only on two points and three times. Why not comparing spatially with model estimates? RESPONSE: We have now introduced two independent data sets to assess the rover performance and this has been a major change to the paper. These independent measures are distributed gravimetric point samples collected during each survey and estimates from the recently operationalised bureau of meteorology water balance model (5km resolution) estimates of root zone soil moisture. These two independent products are compared against both the intensive and broad scale results to demonstrate. Two new figures have been added results new text has been added to results and discussion sections.

Page 8, Line 264-277, Moderate/Major, Provide more details about the point-area regression analysis. It is not completely clear from the text. RESPONSE: this section has been reworded for clarity and the need for future surveys to improve these relationships

has been added.

Page 16, Figure 1, Minor, Provide scale of the figure and indication of the size of the box. RESPONSE: Scale now added to zoom in area. Box dimensions added to caption too.

Based on the comments above I recommend the publication after MODERATE/MAJOR REVISIONS.

[Figure]

[Figure]

**Fig. 1.** Intensive rover survey versus gravimetric soil moisture and AWRA_L model estimates

[Figure]

**Fig. 2.** Broad scale rover survey versus gravimetric soil moisture and AWRA_L model estimates

---

## Short Comment (SC1) · 15 Sep 2017

Dear David McJannet,

thank you for the interesting study using CRNS roving across scales. As an external observer of the interactive discussion, I would like to comment shortly on the discussion about potential road effects. This topic has been the focus in one of our recent research projects in Europe. Our work is currently under review in another journal, but it is already publicly accessible on the arxiv.org pre-print server: https://arxiv.org/abs/1709.04756

Neutron simulations and dedicated experiments indeed show that roads introduce a bias to the neutron counts if the roads are dryer than the surrounding. But as the

climatic conditions at your sites are rather dry and most of your surveys were off-road, I guess the overall impact to your specific case are minor.

The overestimation of field neutrons ranges between factors of 1 to 1.4, depending on field and road moisture content and road width. Just as an example, if your road is of 3 m width and contains approximittely 6% water equivalent (account for lattice, hydrogen fractions, etc.), then your neutron counts are probably biased by factors of 1.03 to 1.13 for field soil moisture conditions of 10% to 30%, respectively. The effect is probably not significant for your April'16 and March'17 surveys, but could have an influence on your wetter campaign days, e.g., June'16.

The good news is that the effect almost vanishes beyond a few meters away from the road. You mentioned in your manuscript that most of your surveys went along paths at the field borders next to the road. Our results suggest that road effects along these off-road tracks are likely to be insignificant compared to the overall heterogeneity of hydrogen pools in the environment.

I hope you find this comment useful and that those recent results could help to support your argumentation in the manuscript.

Regards, Martin Schrön

---

## Referee Comment (RC4) · R. Baatz (Referee) · 19 Sep 2017

General Comments:

The manuscript nicely describes a blueprint for a roving cosmic ray neutron sensor application (CNRS) for remote sensing validations, land surface model validation, and field scale soil moisture retrieval over adequately large farmlands. Conditions for application, and guidelines are outlined in sufficient technical depth. The novelty of this manuscript is the clear presentation of the technical methodology, focus on the purpose, conclusiveness of the experiment and validation of the derived data by static and additional roving CRNS experiments. Hence, the manuscript deserves to be published in HESS subject to revisions which can be easily handled by the authors.

[Figure]

Specific and technical comments:

There are few points which will improve the quality of the paper mostly regarding restructuring of the text and improving clarity of the figures. Although methods and results are mixed at several instances, the manuscript was written fluently and well readable, containing the necessary technical details and contents for reproducible. Along with restructuring, novelties might be marked more strikingly by additional sub headings.

l 42: scale hectometers - rephrase

l 46: remove "better" otherwise better than what?

l 48: I'd suggest to treat land surface modeling separate to remote sensing, and include parameter estimation studies which actually use CRNS already (and are potential cases for rover application at horizontal scale) such as Baatz et al. 2017 "improved land surface model prediction" and Villarreyes et al. 2014 "Inverse modelling of cosmic-ray..." .

l 83: indicate time over which is averaged (monthly average or daily average)

l 86: Stick to one terminology. The authors switch repeatedly between CRNS (this one should be preferred), "cosmic ray soil moisture sensor" and many others throughout the manuscript and headings.

l 93: add citation (e.g. Hawdon et al.)

l 131: isn't air pressure (fp) used to scale to sea level (1013 hPa) instead of an additional scaling factor (fs)? This would avoid using a redundant scaling factor fs.

l 152: This sub-chapter can be restructured mostly to include sections from "Results" but which actually are "Methods". Here, the novelties and blueprint character could be more concise.

l 187: This is not an "additional" part. Now, it is part of this study.

l 197: This is very likely the approach taken by Baatz et al. "An empirical veg.." Eq. 2.

l 209-210: Move to methods

l 214-217: What is remarkable similar? Just the results should be clear enough. Here, the curve-average resulting difference in soil moisture should be also noted, since this is the variable of interest for hydrologists. As it reads now: The interpretation would be that biomass pools are equal. Perhaps, knowing the site conditions, biomass "is basically non-existent".

l219-221: Move to methods. Paragraph reads like the approach described in Baatz et al. 2015.

l 228-232: Move to methods.

l 232-233: This is a result.

l 233-235: Move to methods.

l 249-251: Move to methods or rephrase.

l 259: Please investigate.

l264: "farm property" seems a key words and should be introduced earlier.

l 267-272: Move to methods.

l 275-277: Move to after-results e.g. conclusion or outlook.

l 290ff: to methods.

l 295-299: Link/relate results to driving speed and counting rates.

l 320: replace will with with

Fig. 1: Insert Map of Australia and consider landscape format of the figure.

Fig. 2 and others: Add axis title (Lat/Lon).

Fig. 3: Consider using color bars with 2 colors for b, c, and d. "m ASL" was used in the text, so please use it in the figure as well. Now it is "m AHD".

Fig. 7 and 11: Consider dark Brown- light Yellow or other color bars with 2 colors for neutron counts. Is this already corrected neutron counts? This would be desirable, please indicate. Soil moisture is preferably shown with red-green-blue color bar throughout all plots. The counts shown should be corrected neutron counts. Otherwise, the additional value is not clear. Why are the interpolation patterns for neutron counts not visible in the soil moisture interpolations? I suggest to coarsen the visual representation.

---

## Author Comment (AC7) · 22 Sep 2017

R. Baatz (Referee) r.baatz@fz-juelich.de

THANKS - Thanks for your comments and suggestion Roland - much appreciated. Comments below as RESPONSES (changes will be in revised manuscript)

General Comments: The manuscript nicely describes a blueprint for a roving cosmic ray neutron sensor application (CNRS) for remote sensing validations, land surface model validation, and field scale soil moisture retrieval over adequately large farmlands. Conditions for application, and guidelines are outlined in sufficient technical depth. The

[Figure]

novelty of this manuscript is the clear presentation of the technical methodology, focus on the purpose, conclusiveness of the experiment and validation of the derived data by static and additional roving CRNS experiments. Hence, the manuscript deserves to be published in HESS subject to revisions which can be easily handled by the authors. Specific and technical comments: There are few points which will improve the quality of the paper mostly regarding restructuring of the text and improving clarity of the figures. Although methods and results are mixed at several instances, the manuscript was written fluently and well readable, containing the necessary technical details and contents for reproducible. Along with restructuring, novelties might be marked more strikingly by additional sub headings. l 42: scale hectometers - rephrase RESPONSE: Fixed

l 46: remove "better" otherwise better than what? RESPONSE: Fixed

l 48: I'd suggest to treat land surface modeling separate to remote sensing, and include parameter estimation studies which actually use CRNS already (and are potential cases for rover application at horizontal scale) such as Baatz et al. 2017 "improved land surface model prediction" and Villarreyes et al. 2014 "Inverse modelling of cosmicray..." . RESPONSE: We now treat remote sensing, land surface modelling and parameter estimation studies separately and have included those citations as suggested.

l 83: indicate time over which is averaged (monthly average or daily average) RESPONSE: the time period is daily - fixed

l 86: Stick to one terminology. The authors switch repeatedly between CRNS (this one should be preferred), "cosmic ray soil moisture sensor" and many others throughout the manuscript and headings. RESPONSE: will now use preferred option

l 93: add citation (e.g. Hawdon et al.) RESPONSE: added as suggested

l 131: isn't air pressure (fp) used to scale to sea level (1013 hPa) instead of an additional scaling factor (fs)? This would avoid using a redundant scaling factor fs. RE-

[Figure]

SPONSE: In our case a reference level for fp of sea level is used so in this case the fs is redundant – we leave it in as it may be used in other studies if a different reference elevation is used for air pressure correction (e.g. site average Pressure).

l 152: This sub-chapter can be restructured mostly to include sections from "Results" but which actually are "Methods". Here, the novelties and blueprint character could be more concise. RESPONSE: Many changes made and all method texts now moved to the methods section as suggested.

l 187: This is not an "additional" part. Now, it is part of this study. RESPONSE: correct – reworded

l 197: This is very likely the approach taken by Baatz et al. "An empirical veg.." Eq. 2. RESPONSE: citation added and new text included

l 209-210: Move to methods RESPONSE: Shifted to methods as suggested

l 214-217: What is remarkable similar? Just the results should be clear enough. Here, the curve-average resulting difference in soil moisture should be also noted, since this is the variable of interest for hydrologists. As it reads now: The interpretation would be that biomass pools are equal. Perhaps, knowing the site conditions, biomass "is basically non-existent". RESPONSE: Removed remarkable and reworded. Added the average soil moisture difference to.

L 219-221: Move to methods. Paragraph reads like the approach described in Baatz et al. 2015. RESPONSE: This reference has been included now

l 228-232: Move to methods. RESPONSE: removed from results and covered in methods

l 232-233: This is a result. RESPONSE: agree

l 233-235: Move to methods. RESPONSE: Have left this here with revised wording as the new lattice water product is a result of this study and the result loses its context if

this is not included

l 249-251: Move to methods or rephrase. RESPONSE: removed – scale issue covered in methods

l 259: Please investigate. RESPONSE: we have no way to investigate these differences we can only speculate as to the difference based on local observations. Any point in the rover survey is interpolated using a number of neighbouring points based on the variogram relationships – if there is a very abrupt change in counts in an area it will be smoothed by such an approach.

l264: "farm property" seems a key words and should be introduced earlier. RESPONSE: The concept of farm property and the scale of these is now introduced in the methods

l 267-272: Move to methods. RESPONSE: Moved as suggested

l 275-277: Move to after-results e.g. conclusion or outlook. RESPONSE: Moved to discussion section

l 290ff: to methods. RESPONSE: Moved to methods

l 295-299: Link/relate results to driving speed and counting rates. RESPONSE: Link between driving speed, sample points and counting rates now made clearly

l 320: replace will with with RESPONSE: Fixed

Fig. 1: Insert Map of Australia and consider landscape format of the figure. RESPONSE: Map of Australia now added and landscape format used

Fig. 2 and others: Add axis title (Lat/Lon). RESPONSE: Lat long added to Fig 2, 3, 7, 10 and 12

Fig. 3: Consider using color bars with 2 colors for b, c, and d. "m ASL" was used in the text, so please use it in the figure as well. Now it is "m AHD". RESPONSE: Colour

bars with two colours now used for b,c and d. Changed to m ASL to be consistent with text. Also added LAt/long

Fig. 7 and 11: Consider dark Brown- light Yellow or other color bars with 2 colors for neutron counts. Is this already corrected neutron counts? This would be desirable, please indicate. Soil moisture is preferably shown with red-green-blue color bar throughout all plots. The counts shown should be corrected neutron counts. Otherwise, the additional value is not clear. Why are the interpolation patterns for neutron counts not visible in the soil moisture interpolations? I suggest to coarsen the visual representation. RESPONSE: Dark brown to Yellow is now used throughout for neutron count maps. The neurons are 'corrected' and this has now been made clear in moth figure captions and in the figure itself. Soil moisture is now presented as Red-Green-Blue as suggested in plots 7, and 11 (now 12). The interpolation patterns as the neutron counts are interpolated but are then multiplied by the 90m resolution soil grid data. We have made a point throughout to point put the intended resolution of the intensive and broad-scale surveys.